# Recovery after General Anaesthesia in Adult Horses: A Structured Summary of the Literature

**DOI:** 10.3390/ani11061777

**Published:** 2021-06-14

**Authors:** Miguel Gozalo-Marcilla, Simone Katja Ringer

**Affiliations:** 1Veterinary Clinical Sciences, Easter Bush Campus, The Royal (Dick) School of Veterinary Studies and The Roslin Institute, The University of Edinburgh, Midlothian EH25 9RG, UK; 2Department of Clinical Diagnostics and Services, Vetsuisse Faculty, University of Zürich, 8057 Zürich, Switzerland

**Keywords:** equine, anaesthesia, recovery, general anaesthesia, horse (*Equus caballus*)

## Abstract

**Simple Summary:**

Recovery is the most dangerous phase of general anaesthesia in horses. Numerous publications have reported about this phase, but structured reviews that try to reduce the risk of bias of narrative reviews/expert opinions, focussing on the topic are missing. Therefore, the aim of the present article was to publish the first structured review as a summary of the literature focussing on the recovery phase after general anaesthesia in horses. The objective was to summarise the available literature, taking into account the scientific evidence of the individual studies. A structured approach was followed with two experts in the field independently deciding on article inclusion and its level of scientific evidence. A total number of 444 articles, sorted by topics and classified based on their levels of evidence, were finally included into the present summary. The most important findings were summarised and discussed. The present structured review can be used as a compilation of the publications that, to date, focus on the recovery phase after general anaesthesia in adult horses. This type of review tries to minimise the risk of bias inherent to narrative reviews/expert opinions.

**Abstract:**

Recovery remains the most dangerous phase of general anaesthesia in horses. The objective of this publication was to perform a structured literature review including levels of evidence (LoE) of each study with the keywords “*recovery anaesthesia horse*”, entered at once, in the search browsers PubMed and Web of Science. The two authors independently evaluated each candidate article. A final list with 444 articles was obtained on 5 April 2021, classified as: 41 “*narrative reviews/expert opinions*”, 16 “*retrospective outcome studies*”, 5 “*surveys*”, 59 “*premedication/sedation and induction drugs*”, 27 “*maintenance with inhalant agents*”, 55 “*maintenance with total intravenous anaesthesia (TIVA)*”, 3 “*TIVA versus inhalants*”, 56 “*maintenance with partial intravenous anaesthesia (PIVA)*”, 27 “*other drugs used during maintenance*”, 18 “*drugs before/during recovery*”, 18 “*recovery systems*”, 21 “*respiratory system in recovery*”, 41 “*other factors*”, 51 “*case series/reports*” and 6 “*systems to score recoveries*”. Of them, 167 were LoE 1, 36 LoE 2, 33 LoE 3, 110 LoE 4, 90 LoE 5 and 8 could not be classified based on the available abstract. This review can be used as an up-to-date compilation of the literature about recovery after general anaesthesia in adult horses that tried to minimise the bias inherent to narrative reviews.

## 1. Introduction

Recovery is the phase of general anaesthesia that still implies the highest risk of mortality in horses. The CEPEF2 (Confidential Enquiry of Equine Perioperative Fatalities), the largest multicentre study, reported that one third of all the deaths classified as noncolic deaths, up to 7 days after general anaesthesia, were due to fractures, neuropathies or myopathies related specifically to the recovery phase [1]. Even with the implementation of supervised training programmes in Veterinary Anaesthesia and Analgesia, newer and more sophisticated drugs, protocols and monitoring, deaths related to general anaesthesia in horses still happen, with a big number occurring in recovery. Indeed, “*we still lose horses after anaesthesia to a range of catastrophes that would not occur if the horses were not anaesthetized*” [2]. This is reflected in the tendency of avoiding general anaesthesia whenever possible, moving onto protocols in the standing horse [3]. However, general anaesthesia is still essential in an equine clinic.

Data of Johnston et al., (2002) [1] is now more than 20 years old and there is an obvious need for an update [4], as things have evolved and numbers might have changed as several variables did. Hopefully, results of an ongoing multicentre CEPEF4 study will come up in the following years and will give us more information about the current situation in equine anaesthesia in general, and recovery in particular [5,6]. In the meantime, narrative reviews focussing on the recovery phase do exist that compile many of the studies published up to that date [7,8,9,10]. Since then, a large number of research papers giving us information about different aspects of the recovery phase has been published.

With this background, we aim to publish the first structured review as a summary of the literature focussing on the recovery phase after general anaesthesia. The use of a methodical, comprehensive, transparent and replicable approach minimises the risks of subjectivity and bias of narrative reviews [11,12]. Moreover, we aim to report the levels of evidence (LoE) of each study, an important component of evidence-based medicine that will help the reader to prioritise information [13].

Therefore, our objective was to answer the following questions: (i) How many studies have been published on this topic until today? and (ii) what is the scientific evidence of each study? Finally, based on the information of each publication, (iii) what can be summarised about the anaesthetic recovery phase in adult horses?

## 2. Materials and Methods

A literature search was performed including the three keywords “*recovery anaesthesia horse*”, entered at once, in two search engines commonly used in veterinary anaesthesia: PubMed [https://pubmed.ncbi.nlm.nih.gov (accessed on 22 September 2020)] and Web of Science [https://apps.webofknowledge.com (accessed on 22 September 2020)]. The Web of Science search was by “*topic*” with “*all years (1864–2020)*” as timespan and included “*all databases*” (by default Web of Science Core Collection, BIOSIS Citation Index, Current Context Connect, Data Citation Index, Derwent Innovation Index, KCI—Korean Journal Database, MEDLINE^®^, Russian Science Citation Index, SciELO Citation Index, Zoological Record).

The two authors independently evaluated each candidate article obtained from the two browsers. To obtain as much information as possible, the networks of the University of Zürich (Switzerland) and The University of Edinburgh (United Kingdom) were used. The inclusion criteria for the final list included publications or articles with at least a written abstract in English which included information about the recovery phase from general anaesthesia in adult horses. Truncated abstracts were only included if information regarding the recovery phase was available. Exclusion criteria included studies about foals, donkeys or mules, and candidate articles with no other information about the topic than the title.

During this independent evaluation process, each author classified each candidate article as “*eligible*” or “*not eligible*”. Those classified as “*eligible*” were given an LoE (Table 1) independently by each author and assigned to one of the following categories: “*narrative reviews/expert opinions*”, “*retrospective outcome studies*”, “*surveys*”, “*premedication/sedation and induction drugs”, “maintenance with inhalant agents*”, “*maintenance with total intravenous anaesthesia (TIVA)*”, “*TIVA versus inhalants*”, “*maintenance with partial intravenous anaesthesia (PIVA)*”, “*other drugs used during maintenance*”, “*drugs before/during recovery*”, “*recovery systems*”, “*respiratory system in recovery*”, “*other factors*”, “*case series/reports*” and “*systems to score recoveries*”.

During the following week, the authors pooled their screened results together, created a final inclusion list, classified each study within one of the above categories and provide each with an LoE by consensus. Data were updated on 5 April 2021 before writing the final version of the manuscript.

For this process, the PRISMA-ScR guidelines, used for structured reviews, were followed [14] (Appendix A).

## 3. Results

After evaluation of each article from the initial search by the two authors independently, the final inclusion list according to Section 2 was created and updated on 5 April 2021, with a final number of 458 articles. From them, 14 of the selected articles were excluded as they did not fulfil the inclusion criteria or because the abstract was truncated and, therefore, not assessable, leading to a final list of 444 selected articles (Figure 1).

The total number of the selected studies, the different divisions by categories and the different LoE per division are shown in Table 2. Table 3, Table 4, Table 5, Table 6, Table 7, Table 8, Table 9, Table 10, Table 11, Table 12 and Table 13 show the accepted articles classified by categories and including the LoE of each one [7,8,9,10,15,16,17,18,19,20,21,22,23,24,25,26,27,28,29,30,31,32,33,34,35,36,37,38,39,40,41,42,43,44,45,46,47,48,49,50,51,52,53,54,55,56,57,58,59,60,61,62,63,64,65,66,67,68,69,70,71,72,73,74,75,76,77,78,79,80,81,82,83,84,85,86,87,88,89,90,91,92,93,94,95,96,97,98,99,100,101,102,103,104,105,106,107,108,109,110,111,112,113,114,115,116,117,118,119,120,121,122,123,124,125,126,127,128,129,130,131,132,133,134,135,136,137,138,139,140,141,142,143,144,145,146,147,148,149,150,151,152,153,154,155,156,157,158,159,160,161,162,163,164,165,166,167,168,169,170,171,172,173,174,175,176,177,178,179,180,181,182,183,184,185,186,187,188,189,190,191,192,193,194,195,196,197,198,199,200,201,202,203,204,205,206,207,208,209,210,211,212,213,214,215,216,217,218,219,220,221,222,223,224,225,226,227,228,229,230,231,232,233,234,235,236,237,238,239,240,241,242,243,244,245,246,247,248,249,250,251,252,253,254,255,256,257,258,259,260,261,262,263,264,265,266,267,268,269,270,271,272,273,274,275,276,277,278,279,280,281,282,283,284,285,286,287,288,289,290,291,292,293,294,295,296,297,298,299,300,301,302,303,304,305,306,307,308,309,310,311,312,313,314,315,316,317,318,319,320,321,322,323,324,325,326,327,328,329,330,331,332,333,334,335,336,337,338,339,340,341,342,343,344,345,346,347,348,349,350,351,352,353,354,355,356,357,358,359,360,361,362,363,364,365,366,367,368,369,370,371,372,373,374,375,376,377,378,379,380,381,382,383,384,385,386,387,388,389,390,391,392,393,394,395,396,397,398,399,400,401,402,403,404,405,406,407,408,409,410,411,412,413,414,415,416,417,418,419,420,421,422,423,424,425,426,427,428,429,430,431,432,433,434,435,436,437,438,439,440,441,442,443,444,445,446,447,448,449,450,451,452,453].

## 4. Discussion

This article reports the first structured review that compiles publications that give information about recovery after general anaesthesia in adult horses. For this purpose, two literature database browsers popular in veterinary medicine [454] were used by two investigators working independently. The answers to the formulated questions were: (i) with our search strategy, a total of 444 studies fulfilled the criteria to be included in the final list. (ii) Of them, 167 were classified as LoE 1, 36 as LoE 2, 33 as LoE 3, 110 as LoE 4, 90 as LoE 5 and 8 in which classification was not possible based on the available abstract. Finally, (iii) what can be summarised and discussed about the different factors that affect the anaesthetic recovery phase in adult horses is organised in the subheadings below, referencing to the correspondent table and publications. Scientific evidence was taken into account when summarising the results.

To design our structured literature search, we used a methodical, comprehensive, transparent and replicable approach to fulfil the requirements of a structured literature review. In order to do that, we followed the checklist of the extension for scoping reviews of PRISMA (PRISMA-ScR) [14]. The final aim was to minimise subjectivity and bias inherent to narrative reviews or expert opinions [12]. All the publications were filtered by two experienced researchers in the field that worked independently for data extraction and study quality assessment according to the search strategy. The publications fulfilling the criteria were included in the final list and classified in different tables by topics with a given LoE. Levels of evidence are important components of evidence-based medicine, which should help the reader to prioritise information. However, “*this is not to say that all LoE 4 should be ignored and all LoE 1 accepted as a fact*” [13]. For instance, well-conducted studies with a lower LoE (e.g., LoE 3 retrospective studies with a large number of horses assessing recovery) might be superior and clinically more relevant compared to some studies with a higher LoE (e.g., LoE 1 poorly conducted experimental studies with a low number of research horses assessing recovery). It also needs to be considered that neither blinding nor statistical power are taken into consideration when assigning LoE. Therefore, the reader must always be cautious and critical when interpreting the results of scientific papers.

### 4.1. Narrative Reviews/Expert Opinions

Forty-one narrative reviews/expert opinions were included in our search [2,7,8,9,10,15,16,17,18,19,20,21,22,23,24,25,26,27,28,29,30,31,32,33,34,35,36,37,38,39,40,41,42,43,44,45,46,47,48,49,50]. Four of them gave an overview of the recovery period per se [8,9,10,15], even with explicit titles such as “*What can go wrong?*” [8] and “*Avoiding complications*” [9]. Four reviews focused on specific complications during recovery such as post-anaesthetic myopathy (PAM) [16] and respiratory tract problems [7,17,18]. Eleven publications described the complications of all the phases of general anaesthesia, focussing on the recovery in general [2,19,20,21,22] and in particular cases such as sick patients [23,24,25], orthopaedics [26,27,28] and pregnant mares [29]. The other 21 narrative reviews/expert opinions focussed on the effects of different drugs in anaesthesia with references to the recovery phase [30,31,32,33,34,35,36,37,38,39,40,41,42,43,44,45,46,47,48,49,50].

These publications were written by acknowledged experts in the field. In general, narrative reviews/expert opinions pull many pieces of information together into a readable format and are extremely useful for educational purposes [455]. However, this approach has certain limitations. First, the methodology is often not discussed, the approach is unsystematic and a literature search strategy is commonly missing. Second, narrative reviews/expert opinions rarely attempt to be exhaustive in their inclusion of studies, emphasising mainly key studies that are easily accessible and published in major journals which the author is familiar with, or studies written by the authors. As a consequence, article selection tends to be subjective, lacking explicit criteria for inclusion and exclusion, which could lead to bias [455,456]. All these drawbacks can be avoided by a methodical, comprehensive, transparent and replicable review [12]. Structural/systematic approaches will increase the LoE of the review as well.

### 4.2. Retrospective Outcome Studies

Sixteen studies fell into this category [51,52,53,54,55,56,57,58,59,60,61,62,63,64,65,66]. Most of them are single-centre and give valuable information about mortality, fatalities and outcomes associated to the recovery period. Recovery quality seems to be influenced, amongst others, by the duration of anaesthesia, invasiveness of the procedure, out-of-hours procedures, health status and body mass [51,55,56,58]. Long durations of general anaesthesia are also linked to the occurrence of nerve paralysis [55]. Regarding duration of recovery, a fast recovery does not always imply a good recovery [54]. Moreover, some authors stated that “*the longer it took a horse to stand, the better the recovery quality*” [51]. However, too long recoveries are also not ideal. Prolonged anaesthesia times and low temperatures can detrimentally prolong recovery times [54] and have been associated with the presence of post-anaesthetic colic after non-abdominal procedures [57]. Longer periods of intraoperative hypotension have also been associated with prolonged recoveries [54]. Treatment of hypotension in halothane-anaesthetised horses has been shown to reduce the incidence of severe PAM [51].

Other retrospective outcome studies looked at recovery after specific interventions. Two studies found orthopaedic procedures to be associated with increased mortality [53,58]. With regards to diagnostics, the risks of magnetic resonance imaging (MRI) may not be greater than surgical procedures [65]. However, even when the study was underpowered, 8 of the 350 MRI horses (2.3%) suffered from PAM or neuropathy, compared to only 2 of 229 (0.9%) after surgical procedures. Transient neuropathy seems to be more frequent following MRI of the proximal metatarsal and tarsal structures when compared with other structures, probably because of patient positioning and limb traction [405].

In colic horses, prolonged anaesthetic times were associated with poor recovery qualities, and those with hypoxaemia and hypotension were at higher risk of death and poor recoveries [59]. Especially in colics, correct positioning and cushioning, adequate monitoring during maintenance and assistance for recovery are important [61]. Voulgaris and Hofmeister (2009) suggested to assist recovery with head and tail ropes in colics with hypoxaemia, hypotension, old age, endotoxemia or hypothermia [54]. The discussion about recovery in colic horses is further complemented later on (see subheading “*other factors*”).

In mares anaesthetised for dystocia, low total protein, high temperature and severe dehydration at presentation, prolonged dystocia and intraoperative hypotension increased the probability of peri-anaesthetic death [63]. In that study, receiving a small dose of an alpha2-adrenergic agonist for recovery carried a 9 and 25 times lower risk of death than after alpha2-adrenergic agonist + ketamine and alpha2-adrenergic agonist + ketamine + ketamine, respectively.

For the treatment of bladder stones, standing interventions when possible are suggested to avoid complications of general anaesthesia [64].

Finally, an interesting retrospective study and opinion poll focussed on the use of acepromazine in horses and concluded that there is no justification for a restricted use of acepromazine in intact males compared to geldings and mares [66].

### 4.3. Surveys

Five publications were classified as “*surveys*” [67,68,69,70,71]. Johnston et al., (1995) evidenced that the likelihood of death was increased in longer anaesthesias, in emergency colics, during out of hours, in orthopaedics requiring internal fixation or in mares in the last trimester of pregnancy [67]. Unfortunately, our search did not find the CEPEF2 study which reported the risks of death up to 7 days after general anaesthesia: overall 1.9%, 0.9% in healthy horses and 11.7% in colics [1]. From those classified as noncolic deaths, one third of the deaths was due to cardiac arrest or cardiovascular collapse, one third due to fractures or myopathies and another third due to other complications, e.g., abdominal, respiratory, central nervous system or “*found dead*”. Wohlfender et al., (2015) published the results of an online survey, including private centres and university teaching hospitals [70]. The study described what current practice was six years ago in all the phases of general anaesthesia. Focussing on the recovery, Kästner (2010) published the results of an electronic survey with data from different private and university equine hospitals from several countries [68]. It compiled information about the measures taken to prepare a horse for the recovery phase and the different criteria used to assist recovery or not. The survey conducted by Schrimpf et al., (2011) focussed exclusively on recovery systems used after osteosynthesis, and concluded that head and tail ropes as the most frequently used method to assist recovery in these horses (54%) [69]. A recent online survey collected data concerning current practice of recovering horses and recovery personnel safety [71]. Worldwide practitioners from 22 countries completed 373 questionnaires providing interesting results which are discussed more in detail afterwards (see subheadings “*drugs before/during recovery*”, “*recovery systems*” and “*other factors*”).

As these surveys include information of different sections, we will refer to them several times below.

### 4.4. Premedication/Sedation and Induction Drugs

Fifty-nine studies investigated the possible influence of drugs used during premedication and induction on anaesthetic recovery [72,73,74,75,76,77,78,79,80,81,82,83,84,85,86,87,88,89,90,91,92,93,94,95,96,97,98,99,100,101,102,103,104,105,106,107,108,109,110,111,112,113,114,115,116,117,118,119,120,121,122,123,124,125,126,127,128,129,130]. Seven narrative reviews/expert opinions (Table 3) provided additional information [30,31,32,33,34,35,50].

Acepromazine and different alpha2-adrenergic agonists are the most frequently used tranquilizers/sedatives before induction of anaesthesia in adult horses [70]. Premedication with acepromazine is used frequently as part of a standard protocol [1,66,70]. No large controlled clinical studies comparing the effects of acepromazine on recovery when administered during premedication were found. However, its positive modulatory effects suggest to improve recovery behaviours in horses. Due to its prolonged half-life, the calming effect of acepromazine might still play a role during recovery after anaesthesias of short to intermediate duration [457,458].

With regard to alpha2-adrenergic agonists, only a few studies investigated the influence in the premedication of these drugs on anaesthetic recovery [72,74,75,88,89,122]. Based on these studies, there is minimal evidence that the type of alpha2-adrenergic agonists used during premedication affects quality of recovery. Matthews et al., (1991) reported a tendency to a higher number of attempts needed to stand up when detomidine instead of xylazine was used [122]. Premedication with longer-acting alpha2-adrenergic agonists, such as romifidine, might be superior if no sedation is used during recovery [89].

Together with acepromazine and alpha2-adrenergic agonists, opioids are used in the premedication to enhance sedation and provide analgesia in horses. Their use in premedication might affect recovery from general anaesthesia [73,77,80,82,84,93]. However, if any, only an influence on recovery time and not quality was shown [73,80,83,84,85,93]. But significantly better recoveries were observed when morphine was administered in the already anaesthetised horse, 20 min after induction [220]. Longer recovery times with opioids [73,80,84] might be attributed to a potentiation of the sedative effect of alpha2-adrenergic agonists or due to their analgesic effect, leading to better comfort with less flight instinct [459]. This might also be the reason why horses receiving butorphanol and phenylbutazone presented longer recoveries compared to horses receiving one of the drugs alone [80].

Ketamine is the most frequent agent used to induce general anaesthesia in horses [1,70]. Ketamine is superior to thiopental and the mixture tiletamine/zolazepam regarding duration and quality of recovery [94,98,104,109,110,122]. Thiopental might be still used as an alternative in ocular patients to avoid increases in intraocular pressures [102]. In clinical practice, ketamine is frequently combined with muscle relaxants, mainly diazepam [70], but also midazolam or guaiphenesin, called glyceryl guaiacolate ether (GGE). These drugs smoothen induction and facilitate intubation. However, midazolam might dose dependently affect recovery, and therefore, higher doses should be avoided, especially if short procedures are planned [97,105,106].

Newer induction agents, such as propofol and alfaxalone, could be used for the induction of general anaesthesia too; however, they are not better compared to ketamine considering its influence on recovery [99,101,102,104]. Additionally, its use in adult horses is limited due to their high volumes required and, therefore, increased costs compared with other intravenous agents used for induction of general anaesthesia.

### 4.5. Maintenance with Inhalant Agents

Twenty-seven original studies [131,132,133,134,135,136,137,138,139,140,141,142,143,144,145,146,147,148,149,150,151,152,153,154,155,156,157] were allocated to this category, with six narrative reviews [37,38,39,40,41,42] (Table 3).

Currently, isoflurane [70] and sevoflurane are the most common inhalant agents used in equine anaesthesia, providing smooth, calm recoveries, with minimal differences in recovery times and quality, if any [136,138,142,341].

The search for the ideal inhalant dates from the beginning of last century. In the 1950s, halothane was introduced, avoiding the difficulties and dangers of chloroform and chloral hydrate [460]. Although used for three decades and very popular in the 1980s, halothane led to arterial hypotension, respiratory depression and undesirable long recoveries in horses [461]. Enflurane, a less soluble agent led to faster recoveries but with more shivering and incoordination and did not replace halothane. At that time, Hall stated that “*reports from the USA suggest that another agent, isoflurane (Forane; Ohio Medical) may have advantages over halothane as an anaesthetic for horses*” [461].

Nowadays, halothane is neither manufactured nor used in most of the countries, and isoflurane and sevoflurane are used more often instead. Desflurane might be a good option as well, with quick, rapid recoveries, but it has high-impact atmospheric effects and its use is discouraged [462]. The atmospheric lifetimes for sevoflurane, isoflurane and desflurane are 1.1, 3.2 and 14 years, respectively [463].

### 4.6. Maintenance with TIVA

Fifty-five studies fell into this category, describing different TIVA protocols for maintenance of general anaesthesia and giving information about the recovery phase [158,159,160,161,162,163,164,165,166,167,168,169,170,171,172,173,174,175,176,177,178,179,180,181,182,183,184,185,186,187,188,189,190,191,192,193,194,195,196,197,198,199,200,201,202,203,204,205,206,207,208,209,210,211,212]. Three more compared TIVA techniques versus inhalant agents [213,214,215]. Additionally, five narrative reviews [37,41,47,48,49] (Table 3) provided extra information.

Historically, chloral hydrate and pentobarbital were the main drugs used, with an evolution towards thiopental, commonly combined with GGE [461]. However, recoveries from barbiturates were often prolonged and rough and therefore they were soon replaced by ketamine [178]. Ketamine-based protocols preserve the cardiovascular function better compared to inhalation anaesthesia, as reported by Luna et al., (1996) [215] and McMurphy et al., (2002); this last study was not found in our search [464]. For short interventions (up to 30–60 min) the combination of an alpha2-adrenergic agonists with ketamine provides reliable anaesthesia and acceptable recoveries when applied as repeated boli or infusion [160,179,182,197,208].

The addition of the central muscle relaxant GGE to an alpha2-adrenergic agonists and ketamine is commonly administered as an infusion, the so-called “*triple drip*”, and allows to maintain general anaesthesia for up to 60–90 min in horses. This approach is popular under field and hospital conditions, as it avoids the peaks after boli administrations, it is easy to use and cheap and nonspecific anaesthetic equipment is required. Several studies have reported its safe use, with recoveries being of acceptable duration and quality [51,158,162,172,185,187,188,190,191,194]. However, high doses of GGE can negatively affect quality of recovery, and therefore, its use should be limited in time and total amount [185,194]. Benzodiazepines could be used as central muscle relaxants as well. Midazolam instead of GGE might have a dose-dependent influence on recovery, and therefore, an antagonist may be considered [171,176,201,206,211].

The pharmacological profiles of propofol [465] and alfaxalone [115] in horses, with minimal accumulation and rapid elimination, might allow these drugs to be used for TIVAs longer than 60–90 min, but resultant hypoventilation might limit their use under field conditions [169,173,205]. When using propofol and alfaxalone for TIVA, recoveries range from satisfactory to good and are not inferior to ketamine [167,168,202,204,212]. However, recoveries might be affected dose-dependently, and therefore, neither propofol nor alfaxalone seem to be ideal as sole maintenance agents for prolonged interventions [165,189,203,204,210,212].

Different studies investigated the combination of propofol and alfaxalone with other drugs, such as ketamine, medetomidine, butorphanol, GGE and lidocaine [161,163,169,173,175,192,193,196,198,199,200,205,207,209,210]. As with ketamine, the combination with other drugs and the resulting reduction in total dose of propofol and alfaxalone is advantageous for the recovery phase. Excitation during the early recovery phase has been reported after TIVA based on propofol [195,204] and alfaxalone [174,212]; however, alpha2-adrenergic agonists were not used during recovery in the aforementioned studies. Finally, recovery from TIVA with barbiturates and tiletamine/zolazepam is often prolonged and rough, making its use not recommended for TIVA in horses [159,178,180].

### 4.7. Maintenance with PIVA

Fifty-six original studies, most of them published in the last two decades, fell into this category [216,217,218,219,220,221,222,223,224,225,226,227,228,229,230,231,232,233,234,235,236,237,238,239,240,241,242,243,244,245,246,247,248,249,250,251,252,253,254,255,256,257,258,259,260,261,262,263,264,265,266,267,268,269,270,271]. Five narrative reviews (Table 3) provided additional information [43,44,45,46,47].

Nowadays, the combination of inhalants with intravenous (IV) drugs as constant rate infusions (CRIs) plays a key role in equine anaesthesia and analgesia. The final aims include to reduce the use of inhalants minimising their adverse effects, to maintain adequate surgical conditions with a good intraoperative cardiopulmonary function, followed by a calm, smooth, coordinated recovery [466].

For elective procedures, the use of alpha2-adrenergic agonists CRIs is popular. A stable anaesthetic depth with good haemodynamics is provided, apart from inhalant sparing effects. Good, smooth and calm recoveries follow, which can be of slight longer duration. In studies with unassisted recoveries, more horses after a romifidine CRI stood up with no ataxia at the first attempt when compared with saline [228]. Recoveries after medetomidine PIVA were better compared to lidocaine [222] or S-ketamine [251], and recoveries after dexmedetomidine PIVA were better than after morphine [242]. Recoveries after a dexmedetomidine CRI were better than saline, with longer times to sternal and first attempt to stand [237], and were better than a medetomidine CRI [255]. Comparing alpha2-adrenergic agonists, a recent retrospective study with 78 anaesthetic records showed that PIVA with romifidine provides better recovery qualities than PIVA with detomidine [265]. However, when recovery was assisted with head and tail ropes, recoveries were similar in durations and scores after detomidine or romifidine PIVA [254]. As alpha2-adrenergic agonists increase diuresis and, therefore, urine production, the use of urinary catheters during general anaesthesia is recommended. This will avoid the stimulation by a full bladder and slippery surfaces during recovery.

The use of IV lidocaine is popular in colics, as it has minor cardiovascular effects, i.e., less than medetomidine [222]. It also provides visceral analgesia, promotes gastrointestinal motility, might have anti-inflammatory effects and produces inhalant sparing effects. However, when compared to a medetomidine CRI, recoveries were shorter but worse [222]. Stopping the CRI 20–30 min before the end of surgery is advised to avoid ataxia in the recovery [219]. Adding medetomidine to a lidocaine CRI did not affect cardiovascular function in isoflurane-anaesthetised horses and improved recovery quality compared with lidocaine alone [230].

Ketamine as part of PIVA protocols maintains cardiovascular stability, produces inhalant-sparing effects and is a good extra option when an adequate anaesthetic plane is not achievable with the previous drugs. The main drawbacks when given as a CRI is accumulation that might lead to excitation and rough recoveries. Compared with the racemic ketamine, recoveries after S-ketamine CRIs were better [227]. The excitatory effects that can result in rough recoveries appear to be associated with the R-enantiomer [179,467]. Still, recoveries after a medetomidine CRI were significantly better than S-ketamine [251]. In order to avoid ataxia and uncoordinated recoveries, ketamine protocols should avoid boluses higher than 2 mg/kg, with CRIs no higher than 0.5–1 mg/kg/hr, restraining the duration to 1.5–2 h [466].

Opioids provide analgesia and enhance the sedation of alpha2-adrenergic agonists, but the results of their use for PIVA in horses remain controversial. Morphine as an IV bolus (0.1–0.2 mg/kg) after induction of general anaesthesia did not increase the risk of problems [264] or even improved recoveries [220]. Whereas Chesnel and Clutton (2013) [240] reported uneventful recoveries after morphine CRIs (0.1–0.2 mg/kg/hr), experimental and clinical studies from another research group showed recoveries with excitement [242,245]. A butorphanol CRI did not influence recoveries, but the advantages of its use are limited [229,231,244]. A fentanyl CRI produced highly undesirable, potentially injurious excitatory behaviour [226]. Similar reactions were reported by Thomasy et al., (2006), a study not found in our search, with violent recoveries and hyperthermia [468]. However, when fentanyl and medetomidine CRIs were co-administered, recoveries were without complications [253]. Co-administrations of alpha2-adrenergic agonists could have overwhelmed the potential excitatory effects of opioids [242]. Finally, a remifentanil CRI did not affect recoveries in a randomised, experimental study with 10 horses [247]. No effects on recovery were also observed when both a dexmedetomidine and a remifentanil CRI were co-administered [243].

Apart from using single-drug CRIs, several IV drugs can be used simultaneously with the inhalant agent, for example ketamine and GGE CRIs with halothane [216], medetomidine, ketamine and GGE CRIs with sevoflurane [217] or lidocaine, ketamine, morphine CRIs with isoflurane [233]. References for other combinations are included in Table 6.

### 4.8. Other Drugs Used during Maintenance 

Twenty-seven publications fell into this category [272,273,274,275,276,277,278,279,280,281,282,283,284,285,286,287,288,289,290,291,292,293,294,295,296,297,298]. Our search included nine reports about the use of different loco-regional techniques in the anaesthetised horse, ten about neuromuscular blocking agents (NMBAs) and eight about other drugs, all giving information about the recovery phase.

Nine publications focused on different loco-regional techniques [272,273,274,275,276,277,278,279,280]. In the last few years, a renaissance of loco-regional techniques on standing horses has occurred with the advent of ultrasound guided technology [469,470]. When general anaesthesia is unavoidable, loco-regional techniques can still be used to provide intra- and postoperative analgesia, reduce surgical stimulations and produce muscle relaxation. Local anaesthetics can be used in castrations without implications on the recovery [274,275,276]. Gaesser et al., (2020) reported the safe use of intraarticular mepivacaine for carpal arthroscopy [277]. Epidural xylazine produced halothane sparing effects, suggesting analgesic properties [272], whereas detomidine and morphine, via the same route, provided analgesia after bilateral stifle arthroscopy [273]. However, complications related to epidurals, such as significant ataxia and recumbency, may occur when using local anaesthetics and might have serious consequences on recovery from anaesthesia [280]. A careful selection of drugs and volumes to be injected is mandatory when using this approach.

Apart from the 10 original publications with information about the recovery after general anaesthesia after the use of NMBAs and its reversal agents [281,282,283,284,285,286,287,288,289,290], one narrative review by Martinez (2002) was found in our search [36]. Peripheral muscle relaxants are typically used for ocular [281,285,289], abdominal, orthopaedic, soft tissue procedures [285] or even thoracotomies [285,288]. An early study with the depolarising succinylcholine indicated that, even when ease of recovery was not affected in halothane-anaesthetised horses, muscle damaged was produced by muscle fasciculations [290]. Studies using the non-depolarising muscle relaxants atracurium [283,284,285,286,287], pancuronium [285] and atracurium-vecuronium [289] demonstrated its utility with uneventful recoveries. Apart from edrophonium, Wiese et al., (2014) reported the use of the anticholinesterase inhibitors, neostigmine and physostigmine for reversal or neuromuscular blockade, with superior recovery qualities after physostigmine [282]. Monitoring of neuromuscular function is essential. Residual blockade during recovery from general anaesthesia can lead to respiratory depression and muscle weakness, which could endanger both the horse and personnel [36].

Eight publications reported the effects on recovery of a group of miscellaneous drugs [291,292,293,294,295,296,297,298]. Lee et al., (1998) studied the effects of several cardiovascular stimulant drugs in experimental halothane-anaesthetised ponies [295]. Dobutamine was the most consistent in improving intramuscular blood flow (IMBF), and is most commonly the drug of choice when treating hypotension in horses, without effects on recovery. Phenylephrine did not improve IMBF or cardiac index and two ponies had forelimb lameness during recovery. High rates of dopexamine (10 μg/kg/min) produced sweating and muscular tremors. Additionally, brief lower infusion rates (4 μg/kg/min) produced excitement during the recovery phase with violent shivering [298]. Colic developed as well in two horses within three hours of recovery. Therefore, the clinical use of dopexamine in horses is not recommended.

Antimuscarinics, such as glycopyrrolate [292] or methoctramine [293], doxapram [296], dimethyl sulfoxide [291] and dantrolene [294], do not seem to affect recovery from anaesthesia in horses. Hypertonic solution 7.2%, typically used as a resuscitation fluid in compromised colics, produces an increase in diuresis characterised by numerous micturitions [297]. As they receive large volumes of fluids, the use of urinary catheters is especially recommended in colics to avoid slippery floors during recovery and potential fatalities.

### 4.9. Drugs before/during Recovery 

Eighteen publications fell into this category [299,300,301,302,303,304,305,306,307,308,309,310,311,312,313,314,315,316]. Most of the studies were published after the year 2000 and the main inhalant agent used was isoflurane, followed by sevoflurane.

Surveys showed that the majority of equine anaesthetists currently administer sedatives and/or analgesics before recovery, with alpha2-adrenergic agonists being the most commonly used drugs [68,70,71]. Sedation with alpha2-adrenergic agonists improves the recovery quality [302,303,304], an effect that seems to be dose-dependent [302,311]. There is no clear evidence which alpha2-adrenergic agonist is the best [307,314,315]. However, Bienert et al., (2003) and Woodhouse et al., (2013) showed a benefit of romifidine over xylazine [304,309]. Despite its frequent use, the influence of acepromazine on anaesthetic recovery was only investigated by one study which showed no difference to xylazine (0.15 mg/kg) when administered during the early recovery phase [310]. However, xylazine at 0.15 mg/kg seems to be a rather low dose to optimise recovery [302,311]. The prolongation of general anaesthesia using TIVA (with xylazine/ketamine or xylazine/propofol) after inhalation anaesthesia seems to be superior to no sedation but not to sedation with alpha2-adrenergic agonists [306,308,310]. In mares with dystocias, those receiving for recovery a small dose of an alpha2-adrenergic agonist were at lower risk of death, respectively 9 and 25 times less than those receiving an alpha2-adrenergic agonist + ketamine and alpha2-adrenergic agonist + ketamine + ketamine [63].

A recent study reported the use of flumazenil to counteract the effects of benzodiazepines due to their long plasma half-lives [316]. In that randomised, blinded, crossover, experimental study with six horses, flumazenil shortened recovery times without influencing quality of recovery. However, only a single dose midazolam (0.05 mg/kg IV) was used during induction, and flumazenil was administered 100 min later.

### 4.10. Recovery Systems

Eighteen publications reported different systems to recover horses from general anaesthesia [317,318,319,320,321,322,323,324,325,326,327,328,329,330,331,332,333,334].

Various methods are used to minimise the potential complications associated with recovery [71]. Padded recovery stalls and rope assistance are available in almost every institution. Some hospitals offer additional recovery systems for horses at high risk of catastrophic injury during recovery, such as slings, tilt tables, rapidly inflating-deflating pillows and water-based systems. Data suggest that assistance during recovery is standard in about 40–53% of the clinics [68,70,71]. The survey by de Miguel Garcia et al., (2021) showed that although the majority of equine anaesthetists believe that assistance during recovery decreases the risks, a reduction in fatalities and an improve in quality is still not clearly proven [71].

Head and tail rope assistance is the most frequently used assistance technique [68,69]. Three studies in this review compared the head and tail rope system to unassisted recovery [321,322,323]. Head and tail ropes improved the recovery quality after short elective procedures, which was not shown after long emergency abdominal surgeries [321,322]. The main differences between the two studies were the type of intervention, health status of the horses, duration of anaesthesia and the training level of the person leading the head rope. Nicolaisen et al., (2021) concluded that assistance using head and tail ropes reduces the risk of fatal complications during recovery after emergency abdominal surgery [323]. However, these results should be interpreted with caution, as horses in the assisted group were more frequently sedated for recovery, which would very likely affect the quality of recovery. Larger prospective randomised studies are needed to further investigate the benefits of rope assisted recovery. However, what is clear is that head and tail ropes do not completely prevent the occurrence of fractures during recovery [321,322,323,332], with a suspicion that breeds such as warmblood and in general heavier horses may benefit from this technique [321,323]. Criteria to recover horses with head and tail ropes were hypoxaemia, hypotension, old age, endotoxaemia or hypothermia [54]. In the survey by Wohlfender et al., (2015), criteria included fracture repairs, other orthopaedic procedures, emergency (e.g., colic) surgery or long procedures, bad/very weak general health condition, history a poor recovery, whole limb bandage or cast, intraoperative hypoxaemia or hypothermia, neurological deficits, old horses, ponies, foals and broodmares [70].

Several sling systems have been described to assist recovery after anaesthesia in horses [326,327,329,330,366]. However, most of the studies did not compare sling assistance to other techniques. Only François et al., (2014) compared the Anderson Sling suspension system to unassisted recovery in experimental horses [320]. In general, sling systems seem to be a good alternative for horses at high risk of serious or catastrophic injury during recovery, especially in the absence of a pool recovery system [327,330]. Sling training of a horse before anaesthesia and good sedation during recovery might be beneficial if sling recovery is planned.

Water-based recovery systems can be used to recover horses with increased risk of catastrophic injury [324,325,331,333]. The most frequent complication associated with pool recovery is pulmonary oedema mainly due to the increased extrathoracic hydrostatic pressure when the horse is submerged under water [318]. However, these complications can be avoided or minimised with goal directed anaesthetic management [331]. Disadvantages of pool systems in general are wettening of wounds (hydropool system), size limitations of the raft (pool-raft system), expenses, required man-power and the time commitment.

The air-pillow system requires little manpower and maintenance and recoveries seem to be superior to recovery in a padded recovery box without assistance [319]. However, this statement is based on a single-centre study with horses recovering without sedation after inhalation anaesthesia.

Only one study of the present review investigated the use of a tilt table system for recovery [328]. Although the authors considered the system as useful, 6 of the 36 horses failed to recover on the tilt table.

Finally, the safety of people assisting horses during recovery has been marginally investigated but should be carefully considered [71].

### 4.11. Respiratory System in Recovery

A total of 21 publications fell into this category [335,336,337,338,339,340,341,342,343,344,345,346,347,348,349,350,351,352,353,354,355].

Complications related to the ventilation and oxygenation can happen during general anaesthesia or recovery. Unfortunately, the possible influence of hypoxemia and hypercapnia on anaesthetic recovery in horses remains insufficiently investigated. Rüegg et al., (2016) showed a higher risk for bad recoveries in colic horses if hypoxemia [arterial partial pressure of oxygen (PaO_2_) < 60 mmHg] occurred during anaesthesia [322]. In accordance, shorter recoveries with fewer attempts to stand were observed in colic horses ventilated with constant positive end-expiratory pressure (PEEP) combined with intermittent recruitment manoeuvres (RM) compared to controlled mechanical ventilation (CMV) only [339]. This could have been explained due to the better oxygenation obtained in the PEEP-RM group during anaesthesia. On the other hand, no difference in recovery time or attempts to stand was observed in horses presenting better oxygenation during recovery because of pressure support ventilation during weaning from ventilator [343].

The fraction of inspired oxygen (FiO_2_) settings during general anaesthesia do not seem to affect recovery quality or the PaO_2_ during recovery [340,345,348]. However, horses receiving pulsed delivery of inhaled nitric oxide (NO) during general anaesthesia had better oxygenation and ventilation perfusion (V/Q) matching during recovery [342]. The combination of RM and PEEP in dorsally recumbent horses can improve oxygenation during recovery, but only for the very early recovery phase [339,347].

Different studies agree that hypoxaemia occurs during recovery if oxygen is not supplemented during this phase [345,349,350,351]. It seems that the intrapulmonary shunt produced during general anaesthesia persists in recovery while the horses remain in lateral recumbency, but oxygenation improves once they move to sternal [338,351]. Different techniques and devices tried to improve oxygenation during recovery [343,350,351,353]. Oxygen at 15 L/min via a demand valve or by insufflation into the trachea can help preventing hypoxaemia during recovery [350,351]. Pressure support ventilation during weaning from CMV may help if oxygen is not supplemented during recovery [343]. On the other hand, transporting apnoeic horses to recovery may provide advantages in terms of short-term oxygenation and avoidance of premature emergence from general anaesthesia, compared to horses already weaned from the ventilator before transport [336].

Partial pressure of arterial carbon dioxide (PaCO_2_) during general anaesthesia does not influence the recovery phase [346]. Hypercapnic hyperpnoea aiming to increase alveolar minute ventilation and, therefore, the speed of elimination of inhalant agents, was successful in shortening recovery times, but did not improve quality [341].

As described in several case reports (Table 12) upper airway obstruction is a possible complication during recovery that can be fatal [443,444]. Two narrative reviews cover this topic [7,18]. Nasal phenylephrine can be used to reduce the risk of upper airway obstruction [335].

### 4.12. Other Factors

Forty-one publications reported other factors that might influence recovery [356,357,358,359,360,361,362,363,364,365,366,367,368,369,370,371,372,373,374,375,376,377,378,379,380,381,382,383,384,385,386,387,388,389,390,391,392,393,394,395,396] and seven narrative reviews (Table 3) provided additional information [23,24,25,26,27,28,29].

Twelve original studies [356,357,358,359,360,361,362,363,364,365,366,367] and three narrative reviews focussed on orthopaedic procedures [26,27,28]. One of the major complications associated to recovery after fracture repair is fixation failure and bone refracturing. This has been reported after ulnar [359,360] and third metacarpal/metatarsal condylar fracture repair [358,364]. Other fractures, such as those of the tibia, are challenging mainly due to the size and nature of horses. Still, one successful reduction of an open tibial fracture in an Icelandic horse was reported by Buehler et al., (2011); recovery was performed in a hydropool [366].

Apart from fracture repairs, other orthopaedic procedures should not be underestimated. For instance, bog spavin commonly occurs in Clydesdales, one of the draft breeds that are at higher risks of spinal cord malacia, upper airway obstruction as a consequence of laryngeal hemiplegia, and PAM [28]. In two studies, 1 of 25 and 1 of 18 horses were euthanised because of fractures occurring during anaesthetic recovery from carpal osteochondral fragments surgery and proximal humerus graft collection, respectively [357,363].

The increased risk of recovery in orthopaedic patients was already discussed in Section 4.2. Different authors reported an increased anaesthetic risk in orthopaedic patients [53,58,67]. Johnston et al., (2002) pointed out that not all horses with fractures died due to re-fracture of the operation, but other factors should be considered, such as long durations of surgery and the facts that the animals are stressed, in pain and exhausted/dehydrated as a result of recent, strenuous exercise. This scenario “*contribute to a less than ideal conditions to withstand the further insult of anaesthesia and surgery*” [1].

All this evidence, together with information of the narrative reviews, confirms extreme precautions in these patients. Both Heath (1973) and Auer (2004) [26,27] emphasise the importance of the selected anaesthesia protocols and assistance during recovery. The benefit of assistance during recovery in these patients was also supported by other studies [69,356]. Although not always feasible, procedures on the standing horse are recommended whenever possible [362].

Seven original studies [368,369,370,371,372,373,374] and three narrative reviews focussed exclusively on complications after abdominal surgery, mainly colics [23,24,25]. Anaesthetic-related mortality rates up to seven days after surgery is higher in colics compared with elective procedures (11.7 versus 0.9%) [1]. Nicolaisen et al., (2021) even showed that colic horses are at higher risk to die during recovery when compared to elective and non-nonemergency abdominal procedures [323]. Accordingly, in a further study, recoveries were worse after emergency exploratory laparotomies than those after elective surgeries in dorsal recumbency [371]. Unexpectedly, colics in that study were not more likely to be hypoxaemic. All the horses were mechanically ventilated with intermittent positive pressure ventilation (IPPV). One study stated that mechanical ventilation with IPPV, PEEP and RM is a good option in colic surgery, leading to faster recoveries of similar qualities [339]. As stated in Section 4.2, prolonged anaesthetic times in colics is associated with poor recovery qualities, and those with hypotension and hypoxaemia were at higher risk of death [59]. Assistance of the recovery with head and tail ropes in colics with hypoxaemia, hypotension, old age, endotoxemia or hypothermia has been recommended [54].

Other complications may occur after anaesthesia for abdominal surgery that could increase morbidity/mortality rates [1]. Apart of the complications from surgery itself that reduce survival [62], incisional complications could lead to life-threatening complications [25]. Colic surgeries longer than two hours with hypoxaemic patients had a higher risk of post-operative incisional complications [368,471]. However, this was not found by Robson et al., (2016), who also proposed further prospective studies investigating other factors, such as body temperature, that could play a role in the development of surgical site infections [472]. Both Costa-Farre et al., (2014) and Robson et al., (2016) were not found in our search [471,472]. Poor recoveries after colics are also associated with higher risks of incisional infection, due to either dislodging of the abdominal adhesive dressing/bandage or contamination of the incision by the recovery stall floor [25,473]. The publication by Freeman et al., (2012) was not found in our search [473]. Different measures, such as surgical site skin preparation [370], techniques for incision closure [369], wound protection including abdominal bandages [368] and antibiotic therapy [374], have been investigated to prevent incisional complications.

Other abdominal procedures include emergencies, such as dystocias, in which mares are also at high risk of mortality, 21.5% during the first day post-anaesthesia [63]. One narrative review compiled information about the anaesthesia of mares in the last-term [29].

Elective abdominal procedures include cryptorchidectomies. Laparoscopic approaches under general anaesthesia may be linked to higher incidences of complications, mainly due to increased surgical preparation, surgery and anaesthesia times [372]. Similar to many other surgical procedures, cryptorchidectomies via laparoscopy can be performed in the standing horse whenever possible, avoiding the risks of general anaesthesia [474].

Five studies described complications during the recovery period associated to ocular surgery [375,376,377,378,379]. For these procedures, horses were at higher risk of unsatisfactory recoveries compared to horses undergoing splint bone excision [375]. This could be related to pain from the surgery or even sudden loss of vision in the eye, causing disorientation. The duration of anaesthesia in ocular surgery was identified in two retrospective studies as risk factors [376,377]. In 53 horses, García-López et al., (2009) reported the use of a chain écraseur for enucleation under general anaesthesia [378]. Two out of 53 horses were euthanised after long bone fractures and one horse four days later after severe enterocolitis. No information about assisted or unassisted recovery was given. In a case report by Cary and Hellyer (2002), recovery after general anaesthesia of a horse with a suspected squamous cell carcinoma of the third eyelid was fair, after several attempts without assistance, but with pain and excited [379]. With failure to ventilate, the horse was euthanised, revealing post-mortem a facet fracture C2–C3. As for other procedures, ocular surgery tends towards approaches in the standing horse [475].

Airway surgery such as ventriculectomy or ventriculocordectomy is common in heavy, draft breeds and can be performed under general anaesthesia [380]. With new approaches based on standing sedation and ultrasound guided loco-regional techniques, certain airways procedures may be performed in the standing horse [469], therefore avoiding the risks of recovery from general anaesthesia, especially in those large size breeds.

A negative correlation between body temperature at the end of general anaesthesia and duration of recovery has been described in horses [307]. Body heat is lost during general anaesthesia and also during recovery, especially when the horse is positioned on cold surfaces [381,382]. This is of importance as temperature influences drug metabolism and may, therefore, affect recovery time [476].

Two studies showed that horses gain experience from previous recoveries [310,383]. Both studies were in horses undergoing magnetic resonance imaging that were anaesthetised three times with isoflurane [310] and six times with sevoflurane [383]. Another study classified as “*maintenance with PIVA*” also confirmed that the experience gained during previous recoveries may positively affect future recoveries [252].

One could think that darkening the recovery box would improve recovery qualities. However, this was not found in the prospective, randomised study with 29 horses by Clark-Price et al., (2008) [384] nor in a recent survey [71]. No other publications about this topic were found. Many anaesthetists would probably suspect that noise might influence recovery of horses from anaesthesia, but this has not been sufficiently investigated so far. In a recent survey “*perception of the noise level during the recovery period*” was not associated with worse recovery quality [71].

Three publications focussed on cardiac activity [385,386,387]. The occurrence of atrial fibrillation (AF) or flutter during anaesthetic maintenance or recovery has been described in early studies, similar to the use of IV quinidine to convert AF into normal rhythm [386]. When medical treatment is not possible, transvenous electrical cardioversion (TVEC) can be performed in the anaesthetised horse [477]. Based on a retrospective case series, TVEC under general anaesthesia seems to be relatively safe; however, minor signs of PAM were detected in 6 of the 76 horses, and 1 horse presented facial nerve paralysis [385]. A final study that fell into this category concluded that electrocardiographic variables linked to sympathetic nervous activity cannot predict the quality of recovery [387].

Six studies focussed on metabolic changes that happen during general anaesthesia and that might persist in the recovery phase and the immediate period afterwards [388,389,390,391,392,393]. Although metabolic changes after TIVA [393] and inhalant anaesthesia [388] occur, drugs used currently appear to be relatively safe in this aspect. Increases in lactate, glucose and urea were greater in colics compared to healthy horses [391], and poor recoveries favours increases in blood lactate concentrations [389].

Our search also included three other publications classified as “*miscellaneous*” [394,395,396]. A good care of the eye needs to be taken in the anaesthetised horse, as corneal abrasion/ulceration may occur [395]. The importance of the recovery phase and the potential correlations with different recovery systems still needs to be investigated at this respect. For horses anaesthetised for myelography, the contrast iohexol is a better option than metrizamide, as the latter produces seizures and intensification of preexisting neurological signs and prolongs recoveries [394]. Electroacupuncture appears to not influence recoveries [396]; its use as part of a multimodal analgesic plan in a horse with facial nerve paralysis following general anaesthesia has been described [418].

Finally, information about the effects of different breeds during recovery was intrinsically given in papers under other classifications. Arabian horse breeds were all associated with worse recoveries than Quarter Horses, Thoroughbreds, Warmbloods and others [309]. Draft breeds are associated with higher incidence of idiopathic left laryngeal hemiplegia [380,478] that could lead to upper airway obstruction in the recovery [7,18]. Horses of these heavy weights are also at higher risks of spinal cord malacia and PAM [28]. Linked to this, Laurenza et al., (2020) showed an increase in respiratory and neuromuscular complications with increasing body weight [58].

### 4.13. Case Series/Reports

Fifty-one publications were allocated in this category, 12 case series [397,398,399,400,401,402,403,404,405,406,407,408] and 39 case reports [409,410,411,412,413,414,415,416,417,418,419,420,421,422,423,424,425,426,427,428,429,430,431,432,433,434,435,436,437,438,439,440,441,442,443,444,445,446,447] were included in our final list.

These publications reported cardiac arrests, PAM or neuropathies, fractures, diaphragmatic hernias or even ruptures, post-anaesthetic myelomacias and myelopathies, hyperthermia in recovery, complications related to airway and pulmonary oedema and bladder rupture, among others. As stated by Burns et al., (2011) when classifying the different LoE, “*this is not to say that all LoE 4 should be ignored and all LoE 1 accepted as a fact*” [13]. Indeed, these publications give us very useful information about complications and fatalities that could occur during the recovery period.

### 4.14. Systems to Score Recoveries

Our literature search found six publications describing in detail different systems to score recoveries in horses [448,449,450,451,452,453].

Scarabelli and Rioja (2018) found correlations between systems’ simple descriptive scales (SDS) and composite scoring scales (CSS), the latter giving higher scores [453]. Vettorato et al., (2010) proposed the Edinburgh scoring system, aiming to limit subjective appraisal when differentiating mild vs. hard falls and wall impacts [448]. Compared with a visual analogue scale (VAS) [114], SDS [51] and CSS [134], all four systems were reliable and similarly suitable [448]. Suthers et al., (2011) assessed the reproducibility and repeatability of the SDS, CSS and the Edinburgh scoring system [449]. The three were suitable methods, reasonably reproducible and repeatable, but none was superior to the others. Dichotomous objective scales (DOS) were also reported and compared to VAS and SDS, and could be useful when grading recoveries [451]. Scores from VAS assigned by veterinary anaesthetists, orthopaedic surgeons, equine practitioners and horse owners were similar and independent of gender and experience [450]. The agreement was only affected by gender among horse owners.

Recent results from the system three-axis accelerometry removed bias when compared to SDS or CSS [452]. At the time of publication, the authors claimed that further studies are needed to assess repeatability, sensitivity and specificity.

It is, therefore, concluded that, to date, there is not an ideal, unified system that can be considered the standard to score recoveries.

### 4.15. Limitations

Despite a broad literature search, some relevant publications in the field were unfortunately not found [1,464,466,468,479,480,481]. This could be due to limitations in our methodology. The scope of our search, including all the terms “*recovery anaesthesia horse*” at once, was broad and a more sophisticated approach using different search components (SCs), as used in strict systematic reviews, would have been more appropriate [482]. Splitting the research question into critical SCs and identifying relevant search terms for each SC, for instance Anesthesiology[Mesh] OR Anesthesia[Mesh] with General anesthesia[tiab] OR General anaesthesia[tiab] OR Anesthesia[tiab] OR Anaesthetic[tiab] OR Anesthetic[tiab] OR Anaesthetics[tiab] OR Anaesthesia[tiab] OR Anaesthetised[tiab] OR Anesthetized[tiab] OR, may have avoided the exclusion of certain publications. As an example, whereas including the American English term “*anesthesia*” in “*recovery anesthesia horse*” did not change the number of 468 candidate articles in the PubMed search, the number increased from 727 to 785 in the Web of Science browser. In retrospect, trying to answer a more concrete research question, for example “*Does maintenance with PIVA improve recovery of general anaesthesia in horses?*”, would have help us to deal with a more limited number of references, deepening the discussion of that certain topic. Finally, other relevant studies on the topic were detected in our search for which, unfortunately, no full abstracts were obtained in the browsers and relevant information was missing [483,484]. Therefore, based on our exclusion criteria, these articles were not included in the present review. No systematic approach was followed in order to find articles not detected by the structured reviewing process. However, the present discussion was written by two experts in the field and by doing a literature search as usual to write a scientific manuscript. Therefore, it is unlikely that important articles were overlooked when discussing the results.

Systematic reviews are rare in veterinary anaesthesia, mainly due to the strict requirements [454]. The present work was considered a structured review as the literature search was highly structured in its approach [https://yhec.co.uk/glossary/structured-review/ (accessed on 22 September 2020)] and fulfilled the PRISMA-ScR guidelines [14]. Two different databases were used, with pre-defined inclusion and exclusion criteria. Each article was evaluated and LoEs were assigned by the two authors independently. Additionally, the extracted data of the two authors were compared. Our review encompasses (1) a clearly stated set of objectives with pre-defined eligibility criteria for studies; (2) an explicit, reproducible methodology, and (3) a thorough, objective and reproducible search of a range of sources to identify as many relevant studies as possible. However, our review cannot be classified as a full systematic review, as it does not include (4) an assessment of the validity of the findings for the included studies and (5) a systematic presentation and synthesis of the characteristics and findings of the studies as requested by the Cochrane Handbook for Systematic Reviews of Interventions [https://handbook-5-1.cochrane.org (accessed on 22 September 2020)].

In retrospect, our search could have certainly been improved, mainly as stated above, by limiting the scope of our research question to certain topics and by following a more sophisticated search approach by splitting research questions into critical SCs [482] and including the points 4 and 5 described in the paragraph above to fulfil the criteria of a systematic review. However, our main aim was to provide to the reader with a list of all the publications that, to date, give any information about the recovery phase after general anaesthesia in adult horses. Additionally, we aimed to classify them in a comprehensive way when organising the different tables. In this respect, we believe that, even with a broad topic, we have answered our research questions and reported the first structured review in the topic, attempting to minimise the risk of bias of narrative reviews.

## 5. Conclusions

We report here the first structured review that summarises publications that provide information about the recovery phase after general anaesthesia in adult horses.

General anaesthesia plays a key role in equine practice to perform different diagnostic and surgical procedures. Advances were achieved in the last decades, but complications still happen in the immediate recovery period. With the aim to avoid the risks and complications of general anaesthesia, there is a current tendency towards performing procedures in the standing horse combining sedation protocols with loco-regional techniques.

An update on the current situation is required. In the meantime, this structured review can be used as a compilation of the publications that, to date, focus on the recovery phase after general anaesthesia in adult horses while attempting to minimise the risk of bias of narrative reviews.

## Figures and Tables

**Figure 1 animals-11-01777-f001:**
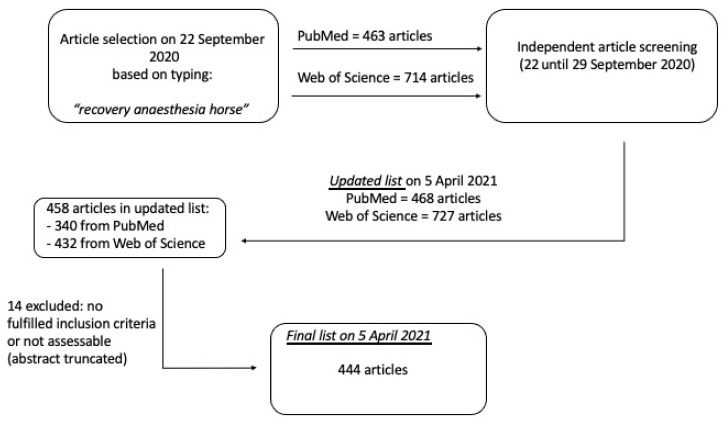
Final list with 444 accepted articles on 5 April 2021. A total of 458 candidate articles were selected. From those, 14 were excluded as they did not fulfil the inclusion criteria or because the abstract was truncated and, therefore, not assessable.

**Table 1 animals-11-01777-t001:** Levels of evidence (LoE) are categorised and reported as adapted from https://www.hydroassoc.org/research-101-levels-of-evidence-in-hydrocephalus-clinical-research-studies/ (accessed on 22 September 2020).

Strength	Level	Design	Randomisation	Control
High	LoE 1	Randomised control trial (RCT)	Yes	Yes
		Meta-analysis of RCT with homogeneous results	No	
	LoE 2	Prospective comparative study (therapeutic)	No	Yes
		Meta-analysis of Level 2 studies or Level 1 studies with inconsistent results	No	
		Prospective cohort study	No	Yes
	LoE 3	Retrospective cohort study	No	Yes
		Case-control study	No	Yes
		Meta-analysis of Level 3 studies	No	
	LoE 4	Case series	No	No
	LoE 5	Case report	No	No
		Expert opinion	No	No
Low		Personal observation	No	No

**Table 2 animals-11-01777-t002:** Final list of 444 selected articles on 5 April 2021, from an updated list of candidate articles of 468 and 727 found with Pubmed and Web of Science, respectively.

Table	Type of Study	Number	Levels of Evidence (LoE)					
			1	2	3	4	5	?
3a	Narrative reviews/expert opinions	41	/	/	/	/	41	/
3b	Retrospective outcome studies	16	/	/	16	/	/	/
3c	Surveys	5	/	5	/	/	/	/
4	Premedication/sedation and induction drugs	59	33	3	2	14	/	7
5a	Maintenance with inhalant agents	27	9	4	/	13	1	/
5b	Maintenance with TIVA	55	19	4	/	32	/	/
5c	TIVA vs. inhalants	3	2	1	/	/	/	/
6	Maintenance with PIVA	56	43	5	2	4	2	/
7a	Loco-regional during maintenance	9	6	/	2	1	/	/
7b	NMBAs during maintenance	10	2	/	/	6	1	1
7c	Other drugs used during maintenance	8	4	1	2	1	/	/
8	Drugs before/during recovery	18	18	/	/	/	/	/
9	Recovery systems	18	5	/	2	9	2	/
10	Respiratory system in recovery	21	15	3	/	3	/	/
11	Other factors	41	8	8	6	15	4	/
12	Case series/reports	51	/	/	/	12	39	/
13	Systems to score recoveries	6	3	2	1	/	/	/
	Total (5 April 2021)	444	167	36	33	110	90	8

Abbreviations in alphabetical order: NMBAs, neuromuscular blocking agents; PIVA, partial intravenous anaesthesia, TIVA, total intravenous anaesthesia. ?: publications in which the study was not able to be classified according to levels of evidence (LoE) based on the available abstract.

**Table 3 animals-11-01777-t003:** Total of: (3a) “*narrative reviews/expert opinions*”, (3b) “*retrospective outcome studies*” and (3c) “*surveys*” including information regarding the recovery phase.

Type of Manuscript	References
(3a) Narrative reviews/expert opinions, LoE = 5	
Recovery period per se	(1) Recovery from anaesthesia in horses [15] (2) Recovery from anaesthesia in horses 1. What can go wrong? [8] (3) Recovery from anaesthesia in horses 2. Avoiding complications [9] (4) Recovery of horses from anesthesia [10]
Postanaesthetic myopathy	(5) Equine postanesthetic myopathy—An update [16]
Pulmonary function GA	(6) Pulmonary function in the horse during anaesthesia: A review [17]
Upper airway during recovery	(7) Post-anaesthetic pulmonary oedema in horses: A review [7] (8) Mitigating the risk of airway obstruction during recovery from anaesthesia: The way is far from clear [18]
Complications during GA	(9) Complications in equine anesthesia [19] (10) Equine anaesthesia-associated mortality: where are we now? [2] (11) Anaesthesia-related complications in horses—results of the last few years [20] (12) Editorial: Anesthetic risk and complications in veterinary medicine [21] (13) Mortality and morbidity in equine anaesthesia [22]
Complications during GA in the clinically ill horse	(14) Anesthesia in horses with colic syndrome: Analysis of 48 cases and literature review [23] (15) Anesthesia of the critically ill equine patient [24] (16) Incisional infections associated with ventral midline celiotomy in horses [25]
Complications during GA in orthopaedics	(17) Anesthetic management and recovery of large orthopedic patients [26] (18) Fracture fixation in horses: Recent developments in implants, management and recovery—A rewiew [27] (19) Bog spavin: Recognising the problem is the first step towards recovery [28]
Complications during GA in late-term mares	(20) Anesthesia and sedation for late-term mares [29]
Effects of α_2_-agonists	(21) Pre-anesthetic medication in the horse part IV. Sedative-hypnotics and drug mixtures [30] (22) The responses to the use of detomidine (Domosedan) in the horse [31] (23) Use of the alpha-2 agonists xylazine and detomidine in the perianaesthetic period in the horse [32] (24) Is there a place for dexmedetomidine in equine anaesthesia and analgesia? A systematic review (2005–2017) [33]
Effects of other IV drugs	(25) Sedation in equine practice—Indications and choice of the methods [34] (26) Clinical insights: Equine anaesthesia and analgesia [35]
Effects of NMBAs	(27) Neuromuscular blocking agents [36]
Effects of inhalant agents	(28*) General anesthesia in pleasure horses [37] (29) Use of halothane and isoflurane in the horse [38] (30) Isoflurane as an inhalational anesthetic agent in clinical practice [39] (31) Desflurane and sevoflurane. New volatile anesthetic agents [40] (32*) Anaesthesia for minor surgical procedures in the horse [41] (33) Recent advances in inhalation anesthesia [42]
Effects of balanced anaesthesia and PIVA (or SIVA)	(34) Use of supplemental intravenous anaesthesia/analgesia in horses [43] (35) Balanced anesthesia in horses [44] (36) Partial intravenous anaesthesia in the horse: A review of intravenous agents used to supplement equine inhalation anaesthesia. Part 1: Lidocaine and ketamine [45] (37) Partial intravenous anaesthesia in the horse: A review of intravenous agents used to supplement equine inhalation anaesthesia. Part 2: Opioids and alpha-2 adrenoceptor agonists [46] (38*) Total and partial intravenous anaesthesia of horses [47]
Effects of TIVA	(28*) General anesthesia in pleasure horses [37] (32*) Anaesthesia for minor surgical procedures in the horse [41] (39) Field anesthetic techniques for use in horses [48] (40) Total intravenous anesthesia in horses [49] (38*) Total and partial intravenous anaesthesia of horses [47]
Effects of agents for pain management	(41) Perioperative pain management in horses [50]
Total narrative reviews/expert opinions = 41	
(3b) Retrospective outcome studies, LoE = 3	
Mortality, fatalities and outcome associated to GA in general	(1) Factors influencing the outcome of equine anaesthesia: A review of 1314 cases [51] (2) Clinical problems during inhalatory anaesthesia in horses. A review of 114 cases [52] (3) Equine perioperative fatalities associated with general anaesthesia at a private practice—a retrospective case series [53] (4) Multivariate analysis of factors associated with post-anesthetic times to standing in isoflurane-anesthetized horses: 381 cases [54] (5) Risk of general anesthesia in horses—A retrospective study on 1.989 cases [55] (6) Twenty years later: A single-centre, repeat retrospective analysis of equine perioperative mortality and investigation of recovery quality [56] (7) Time-related changes in post-operative equine morbidity: A single-centre study [57] (8) Risk factors of anesthesia-related mortality and morbidity in one equine hospital: A retrospective study on 1161 cases undergoing elective or emergency surgeries [58]
Mortality, fatalities and outcome associated to GA in colics	(9) A retrospective survey of anaesthesia in horses with colic [59] (10) Evaluation of exploratory laparotomy in young horses: 102 cases (1987–1992) [60] (11) Anaesthesia related problems in equine colic surgery: A retrospective study from 1995 until 1999 [61] (12) Large colon resection and anastomosis in horses: 52 cases (1996–2006) [62]
Mortality, fatalities and outcome associated to GA for dystocias	(13) Perioperative risk factors for mortality and length of hospitalization in mares with dystocia undergoing general anesthesia: A retrospective study [63]
Mortality, fatalities and outcome associated to GA for cystoliths removal	(14) Cystoliths in the horse—a retrospective study [64]
Post-anaesthetic myopathy/neuropathy in horses undergoing MRI	(15) Post anaesthetic myopathy/neuropathy in horses undergoing magnetic resonance imaging compared to horses undergoing surgery [65]
Use of ACP and its effects on the recovery	(16) Contemporary use of acepromazine in the anaesthetic management of male horses and ponies: A retrospective study and opinion poll [66]
Total retrospective outcome studies = 16	
(3c) Surveys, LoE = 2	
Mortality, fatalities and outcome associated to GA in general	(1) Confidential enquiry of perioperative equine fatalities (CEPEF-1): Preliminary results [67]
Electronic/hard copy questionnaires about assisted recovery	(2) How to manage recovery from anaesthesia in the horse—to assist or not to assist? [68] (3) Relevance of assisting horses in recovery after general anesthesia for osteosynthesis? [69]
Online survey evaluating practice in equine anaesthesia	(4) International online survey to assess current practice in equine anaesthesia [70]
Online survey evaluating current practice of recovering horses from GA	(5) Questionnaire on the process of recovering horses from general anesthesia and associated personnel injury in equine practice [71]
Total surveys = 5	

Abbreviations in alphabetical order: ACP, acepromazine; GA, general anaesthesia; IV, intravenous; LoE, level of evidence; MRI, magnetic resonance imaging; NMBAs, neuromuscular blocking agents; PIVA, partial intravenous anaesthesia; SIVA, supplemental intravenous anaesthesia; TIVA, total intravenous anaesthesia. *: “*narrative reviews/expert opinions*” present in two different categories within the same section.

**Table 4 animals-11-01777-t004:** Total of publications giving information about effects of (4a) “*premedication/sedation*”, (4b) “*induction drugs*” and (4c) “*sedation and induction drugs*” on the recovery phase.

LoE and Topic	References
(4a) LoE premedication/sedation	
1	(1) A comparison of romifidine and xylazine when used with diazepam/ketamine for short duration anesthesia in the horse [72] (2) Effects of additional premedication on romifidine and ketamine anaesthesia in horses [73] (3) Comparison of detomidine and romifidine as premedicants before ketamine and halothane anesthesia in horses undergoing elective surgery [74] (4) Clinical comparison of xylazine and medetomidine for premedication of horses [75] (5) Comparison of morphine and butorphanol as pre-anaesthetic agents in combination with romifidine for field castration in ponies [76] (6) The influence of butorphanol dose on characteristics of xylazine-butorphanol-propofol anesthesia in horses at altitude [77] (7) Dissociative anaesthesia during field and hospital conditions for castration of colts [78] (8) Influence of the combination of butorphanol and detomidine within premedication on the preoperative sedations score, the intraoperative cardiovascular situation and the early recovery period in horses [79] (9) Analgesic effects of butorphanol tartrate and phenylbutazone administered alone and in combination in young horses undergoing routine castration [80] (10) Comparison of acepromazine, midazolam and xylazine as preanaesthetics to ketamine-isoflurane anaesthesia in horses [81] (11) Buprenorphine provides better anaesthetic conditions than butorphanol for field castration in ponies: Results of a randomised clinical trial [82] (12) Effect of pre- and postoperative phenylbutazone and morphine administration on the breathing response to skin Incision, recovery quality, behavior, and cardiorespiratory variables in horses undergoing fetlock arthroscopy: A pilot study [83] (13) Analgesic and adjunct actions of nalbuphine hydrochloride in xylazine or xylazine and acepromazine premedicated horses [84] (14) Preemptive analgesia, including morphine, does not affect recovery quality and times in either pain-free horses or horses undergoing orchiectomy [85] (15) A preliminary study on effects of subanesthetic doses of preemptive ketamine given prior to premedication on total intravenous anesthesia for short- to medium-term surgical procedures in horses [86] (16) Effect of methadone combined with acepromazine or detomidine on sedation and dissociative anesthesia in healthy horses [87] (17) Comparison of xylazine and detomidine in combination with midazolam/ketamine for field castration in Quarter Horses [88]
2	(18) Comparison of romifidine and xylazine as premedicants before general anaesthesia in horses regarding the postsurgical recovery period [89]
3	/
4	(19) Evaluation of xylazine as a sedative and preanesthetic agent in horses [90] (20) A field trial of ketamine anaesthesia in the horse [91] (21) Clinical evaluation of romifidine/ketamine/halothane anaesthesia in horses [92]
5	/
?	(22) Use of a propionylpromazine and meperidine combination in thiopental sodium anesthesia in horses [93]
Total premedication/sedation publications = 22	
(4b) LoE induction drugs	
1	(1) A clinical trial of three anaesthetic regimens for the castration of ponies [94] (2) A comparison of xylazine-diazepam-ketamine and xylazine-guaifenesin-ketamine in equine anesthesia [95] (3) Evaluation of propofol as a general anesthetic for horses [96] (4) Comparison of thiopentone/guaifenesin, ketamine/guaifenesin and ketamine/midazolam for the induction of horses to be anaesthetised with isoflurane [97] (5) Behavioral responses following eight anesthetic induction protocols in horses [98] (6) Evaluation of different doses of propofol in xylazine pre-medicated horses [99] (7) Comparison of high (5%) and low (1%) concentrations of micellar microemulsion propofol formulations with a standard (1%) lipid emulsion in horses [100] (8) Alfaxalone compared with ketamine for induction of anaesthesia in horses following xylazine and guaifenesin [101] (9) Effects of ketamine, propofol, or thiopental administration on intraocular pressure and qualities of induction of and recovery from anesthesia in horses [102] (10) Comparison of midazolam and diazepam as co-induction agents with ketamine for anaesthesia in sedated ponies undergoing field castration [103] (11) Comparison of alfaxalone, ketamine and thiopental for anaesthetic induction and recovery in Thoroughbred horses premedicated with medetomidine and midazolam [104] (12) Recovery of horses from general anesthesia after induction with propofol and ketamine versus midazolam and ketamine [105] (13) Evaluation of the use of midazolam as a co-induction agent with ketamine for anaesthesia in sedated ponies undergoing field castration [106] (14) A comparison of two ketamine doses for field anaesthesia in horses undergoing castration [107]
2	(15*) Methohexital sodium anesthesia in the horse (Trial 1) [108] (16) Intravenous anaesthesia in horses after xylazine premedication [109]
3	(17) A retrospective comparison of induction with thiopental/guaifenesin and propofol/ketamine in Thoroughbred racehorses anesthetized with sevoflurane and medetomidine during arthroscopic surgery [110]
4	(15*) Methohexital sodium anesthesia in the horse (Trials 2 and 3) [108] (18) Effects of Saffan administered intravenously in the horse [111] (19) Observations on the use of glyceryl guaiacolate as an adjunct to general anaesthesia in horses [112] (20) Midazolam and ketamine induction before halothane anaesthesia in ponies: Cardiorespiratory, endocrine and metabolic changes [113] (21) Cardiorespiratory effects of a tiletamine/zolazepam-ketamine-detomidine combination in horses [114] (22) The pharmacokinetics and pharmacodynamics of the injectable anaesthetic alfaxalone in the horse [115] (23) Short-term general anesthesia with tiletamine/zolazepam in horses sedated with medetomidine for castration under field conditions [116]
5	/
?	(24) Thiopentone (pentothal sodium) as a general anaesthetic in the horse [117] (25) Glyceryl guaiacolate in equine anesthesia [118] (26) Pharmacokinetics of ketamine in the horse [119]
Total induction drugs publications = 26	
(4c) LoE sedation and induction drugs	
1	(1) Evaluation of propofol for general anesthesia in premedicated horses [120] (2) Comparison of detomidine/ketamine and guaiphenesin/thiopentone for induction of anaesthesia in horses maintained with halothane [121]
2	(3) A comparison of injectable anesthetic combinations in horses [122]
3	(4) Propofol with ketamine following sedation with xylazine for routine induction of general anaesthesia in horses [123]
4	(5) Clinical trial of xylazine with ketamine in equine anaesthesia [124] (6) Equine castration: A practical approach [125] (7) Clinical observations during induction and recovery of xylazine-midazolam- propofol anesthesia in horses [126] (8) Clinical evaluation of detomidine-butorphanol-guaifenesin-ketamine as short term TIVA in Spiti ponies [127]
5	/
?	(9) Evaluation of xylazine and ketamine hydrochloride for anesthesia in horses [128] (10) Evaluation of xylazine, guaifenesin, and ketamine hydrochloride for restraint in horses [129] (11) Ketamine, Telazol, xylazine and detomidine. A comparative anesthetic drug combinations study in ponies [130]
Total sedation and induction drugs publications = 11	

LoE: level of evidence. ?: publications in which the study was not able to be classified according to LoE based on the available abstract. *: articles containing more than one trial/part with different LoE were classified based on the lowest LoE.

**Table 5 animals-11-01777-t005:** Total of publications giving information about the effects of (5a) “maintenance with inhalant agents”, (5b) “maintenance with total intravenous anaesthesia (TIVA)” and (5c) “maintenance with TIVA versus inhalant agents” on the recovery phase.

LoE and Topic	References
(5a) LoE inhalant agents	
1	(1) Clinical anaesthesia in the horse: Comparison of enflurane and halothane [131] (2) Actions of isoflurane and halothane in pregnant mares [132] (3) Comparison of recoveries from halothane vs isoflurane anesthesia in horses [133] (4) The recovery of horses from inhalant anesthesia: A comparison of halothane and isoflurane [134] (5) Is isoflurane safer than halothane in equine anaesthesia? Results from a prospective multicentre randomised controlled trial [135] (6) Differences in need for hemodynamic support in horses anesthetized with sevoflurane as compared to isoflurane [136] (7) Comparison of hemodynamic, clinicopathologic, and gastrointestinal motility effects and recovery characteristics of anesthesia with isoflurane and halothane in horses undergoing arthroscopic surgery [137] (8) A comparison of recovery times and characteristics with sevoflurane and isoflurane anaesthesia in horses undergoing magnetic resonance imaging [138] (9) Desflurane and sevoflurane elimination kinetics and recovery quality in horses [139]
2	(10) Recovery from anaesthesia in ponies: A comparative study of the effects of isoflurane, enflurane, methoxyflurane and halothane [140] (11) Isoflurane anesthesia for equine colic surgery. Comparison with halothane anesthesia [141] (12) Cardiorespiratory effects of sevoflurane, isoflurane, and halothane anesthesia in horses [142] (13) Clinical comparison of medetomidine with isoflurane or sevoflurane for anesthesia in horses [143]
3	/
4	(14) A study of the use of methoxyflurane general anesthesia in the horse [144] (15) Influence of a clinical anaesthesia-technique (premedication with tranquillizers and atropine, induction with chloralhydrate, maintenance with halothane in a closed circle system) on liver function tests in the horse [145] (16) Clinical experiences with isoflurane in dogs and horses [146] (17) Determination of the minimum alveolar concentration (MAC) and physical response to sevoflurane inhalation in horses [147] (18) Sevoflurane and oxygen anaesthesia following administration of atropine-xylazine-guaifenesin-thiopental in spontaneously breathing horses [148] (19) Cardiovascular and pulmonary effects of sevoflurane anesthesia in horses [149] (20) Anesthetic potency of desflurane in the horse: Determination of the minimum alveolar concentration [150] (21) Maintenance of anaesthesia with sevoflurane and oxygen in mechanically-ventilated horses subjected to exploratory laparotomy treated with intra- and post operative anaesthetic adjuncts [151] (22) Perioperative plasma cortisol concentration in the horse [152] (23) Romifidine-ketamine-halothane anesthesia in horses [153] (24) Anesthetic management with sevoflurane and oxygen for orthopedic surgeries in racehorses [154] (25) Anesthesia in Caspian ponies [155] (26) Validation of the bispectral index as an indicator of anesthetic depth in Thoroughbred horses anesthetized with sevoflurane [156]
5	(27) The modification and performance of a large animal anesthesia machine (Tafonius^®^) in order to deliver Xenon to a horse [157]
Total inhalant agents publications = 27	
(5b) LoE TIVA	
1	(1) Guaifenesin-ketamine-xylazine anesthesia for castration in ponies—a comparative study with two different doses of ketamine [158] (2) Comparison of four drug combinations for total intravenous anesthesia of horses undergoing surgical removal of an abdominal testis [159] (3) Evaluation of xylazine and ketamine for total intravenous anesthesia in horses [160] (4) Evaluation of total intravenous anesthesia with propofol or ketamine-medetomidine-propofol combination in horses [161] (5) Comparison of 3 total intravenous anesthetic infusion combinations in adult horses [162] (6) Evaluation of cardiovascular effects of total intravenous anesthesia with propofol or a combination of ketamine-medetomidine-propofol in horses [163] (7) Comparison of ketamine and S(+)-ketamine, with romifidine and diazepam, for total intravenous anesthesia in horses [164] (8) Anaesthetic and cardiorespiratory effects of propofol at 10% for induction and 1% for maintenance of anaesthesia in horses [165] (9) Short-term anaesthesia with xylazine, diazepam/ketamine for castration in horses under field conditions: Use of intravenous lidocaine [166] (10) Anaesthetic evaluation of ketamine/propofol in acepromazine- xylazine premedicated horses [167] (11) Comparison of ketamine and alfaxalone for induction and maintenance of anaesthesia in ponies undergoing castration [168] (12) Anesthetic and cardiorespiratory effects of propofol, medetomidine, lidocaine and butorphanol total intravenous anesthesia in horses [169] (13) Cardiorespiratory and antinociceptive effects of two different doses of lidocaine administered to horses during a constant intravenous infusion of xylazine and ketamine [170] (14) Effects of dexmedetomidine and xylazine on cardiovascular function during total intravenous anaesthesia with midazolam and ketamine and recovery quality and duration in horses [171] (15) Continuous maintenance anaesthesia using guaifenesin or diazepam combined with xylazine and ketamine in horses [172] (16) Evaluation of total intravenous anesthesia with propofol-guaifenesin-medetomidine and alfaxalone-guaifenesin-medetomidine in Thoroughbred horses undergoing castration [173] (17) Alfaxalone for maintenance of anaesthesia in ponies undergoing field castration: Continuous infusion compared with intravenous boluses [174] (18) Cardiopulmonary effects and recovery characteristics of horses anesthetized with xylazine-ketamine with midazolam or propofol [175] (19) Total intravenous anaesthesia with ketamine, medetomidine and guaifenesin compared with ketamine, medetomidine and midazolam in young horses anaesthetised for computerised tomography [176]
2	(20) Total intravenous anaesthesia in the horse with propofol [177] (21) The stress response to anaesthesia in ponies: Barbiturate anaesthesia [178] (22) Intravenous anaesthesia in hoses: Racemic ketamine versus S-(+)-ketamine [179] (23) Tiletamine-zolazepam anesthesia in horses: Repeated dose versus continuous infusion [180]
3	/
4	(24) Anesthesia by injection of xylazine, ketamine and the benzodiazepine derivative climazolam and the use of the benzodiazepine antagonist Ro 15–3505 [181] (25) Prolongation of anesthesia with xylazine, ketamine, and guaifenesin in horses: 64 cases (1986–1989) [182] (26) Clinical evaluation of an infusion of xylazine, guaifenesin and ketamine for maintenance of anaesthesia in horses [183] (27) A case report on the use of guaifenesin-ketamine-xylazine anesthesia for equine dystocia [184] (28) Total intravenous anaesthesia in ponies using detomidine, ketamine and guaiphenesin: Pharmacokinetics, cardiopulmonary and endocrine effects [185] (29) Physiologic effects of anesthesia induced and maintained by intravenous administration of a climazolam-ketamine combination in ponies premedicated with acepromazine and xylazine [186] (30) Romifidine, ketamine and guaiphenesin continual infusion anaesthesia: Some experiences of its use under field conditions [187] (31) Guaifenesin-ketamine-detomidine anesthesia for castration of ponies [188] (32) Detomidine-propofol anesthesia for abdominal surgery in horses [189] (33) Investigations into injection anesthesia (TIVA) of the horse with ketamine/guaifenesin/xylazin: Experiences with computerized pump infusion [190] (34) Anaesthetic compound and its application in general anaesthesia of horses [191] (35) Cardiopulmonary effects of prolonged anesthesia via propofol-medetomidine infusion in ponies [192] (36) Infusion of a combination of propofol and medetomidine for long-term anesthesia in ponies [193] (37) Intravenous anaesthesia using detomidine, ketamine and guaiphenesin for laparotomy in pregnant pony mares [194] (38) Propofol anaesthesia for surgery in late gestation pony mares [195] (39) Assessment of a medetomidine/propofol total intravenous anaesthesia (TIVA) for clinical anaesthesia in equidae [196] (40) Practical experiences and clinical parameters in a xylazine-ketamine-anaesthesia of horses [197] (41) Medetomidine-ketamine anaesthesia induction followed by medetomidine-propofol in ponies: Infusion rates and cardiopulmonary side effects [198] (42) Propofol-ketamine anesthesia for internal fixation of fractures in racehorses [199] (43) Total intravenous anaesthesia in horses using medetomidine and propofol [200] (44) Anesthetic and cardiopulmonary effects of total intravenous anesthesia using a midazolam, ketamine and medetomidine drug combination in horses [201] (45) Alfaxalone in cyclodextrin for induction and maintenance of anaesthesia in ponies undergoing field castration [202] (46) Pharmacokinetic profile in relation to anaesthesia characteristics after a 5% micellar microemulsion of propofol in the horse [203] (47) Evaluation of cardiovascular, respiratory and biochemical effects, and anesthetic induction and recovery behavior in horses anesthetized with a 5% micellar microemulsion propofol formulation [204] (48) Clinical evaluation of total intravenous anesthesia using a combination of propofol and medetomidine following anesthesia induction with medetomidine, guaifenesin and propofol for castration in Thoroughbred horses [205] (49) Evaluation of a midazolam-ketamine-xylazine infusion for total intravenous anesthesia in horses [206] (50) Alfaxalone and medetomidine intravenous infusion to maintain anaesthesia in colts undergoing field castration [207] (51) Clinical and pharmacokinetic evaluation of S-ketamine for intravenous general anaesthesia in horses undergoing field castration [208] (52) Cardiovascular effects of total intravenous anesthesia using ketamine-medetomidine-propofol (KMP-TIVA) in horses undergoing surgery [209] (53) Cardiorespiratory and anesthetic effects of combined alfaxalone, butorphanol, and medetomidine in Thoroughbred horses [210] (54) Total intravenous anesthesia using a midazolam-ketamine-xylazine infusion in horses: 46 cases (2011–2014) [211] (55) Alfaxalone for total intravenous anaesthesia in horses [212]
5	/
Total TIVA publications = 55	
(5c) LoE TIVA vs. inhalant agents	
1	(1) Cardiovascular effects of surgical castration during anaesthesia maintained with halothane or infusion of detomidine, ketamine and guaifenesin in ponies [213] (2) Cardiopulmonary effects of dexmedetomidine and ketamine infusions with either propofol infusion or isoflurane for anesthesia in horses [214]
2	(3) Cardiorespiratory, endocrine and metabolic changes in ponies undergoing intravenous or inhalation anaesthesia [215]
Total TIVA versus inhalants publications = 3	

LoE: level of evidence. TIVA: total intravenous anaesthesia.

**Table 6 animals-11-01777-t006:** Total of publications giving information about the effects of “*maintenance with partial intravenous anaesthesia (PIVA)*” on the recovery phase.

LoE	References
1	(1) Anaesthesia in horses using halothane and intravenous ketamine-guaiphenesin: A clinical study [216] (2) Infusion of guaifenesin, ketamine, and medetomidine in combination with inhalation of sevoflurane versus inhalation of sevoflurane alone for anesthesia of horses [217] (3) Interactions of morphine and isoflurane in horses [218] (4) Effect of a constant rate infusion of lidocaine on the quality of recovery from sevoflurane or isoflurane general anaesthesia in horses [219] (5) Morphine administration in horses anaesthetized for upper respiratory tract surgery [220] (6) Combined anaesthesia with isoflurane and an infusion of a mixture of ketamine, midazolam and one of three α2-adrenoreceptor agonists for castration in horses [221] (7) A clinical comparison of two anaesthetic protocols using lidocaine or medetomidine in horses [222] (8) The effects of morphine on the recovery of horses from halothane anaesthesia [223] (9) Clinical evaluation of ketamine and lidocaine intravenous infusions to reduce isoflurane requirements in horses under general anaesthesia [224] (10) A study of cardiovascular function under controlled and spontaneous ventilation in isoflurane-medetomidine anaesthetized horses [225] (11) Effects of high plasma fentanyl concentrations on minimum alveolar concentration of isoflurane in horses [226] (12) Evaluation of anesthesia recovery quality after low-dose racemic or S-ketamine infusions during anesthesia with isoflurane in horses [227] (13) Romifidine as a constant rate infusion in isoflurane anaesthetized horses: A clinical study [228] (14) Effects of adding butorphanol to a balanced anaesthesia protocol during arthroscopic surgery in horses [229] (15) Comparison of cardiovascular function and quality of recovery in isoflurane-anaesthetised horses administered a constant rate infusion of lidocaine or lidocaine and medetomidine during elective surgery [230] (16) A clinical study on the effect in horses during medetomidine-isoflurane anaesthesia, of butorphanol constant rate infusion on isoflurane requirements, on cardiopulmonary function and on recovery characteristics [231] (17) Effects of a constant rate infusion of detomidine on cardiovascular function, isoflurane requirements and recovery quality in horses [232] (18) Effects of constant rate infusion of lidocaine and ketamine, with or without morphine, on isoflurane MAC in horses [233] (19) Comparison of the cardiovascular effects of equipotent anesthetic doses of sevoflurane alone and sevoflurane plus an intravenous infusion of lidocaine in horses [234] (20) Comparison of the effects of xylazine bolus versus medetomidine constant rate infusion on the stress response, urine production, and anesthetic recovery characteristics in horses anesthetized with isoflurane [235] (21) Medetomidine continuous rate intravenous infusion in horses in which surgical anaesthesia is maintained with isoflurane and intravenous infusions of lidocaine and ketamine [236] (22) Influence of a constant rate infusion of dexmedetomidine on cardiopulmonary function and recovery quality in isoflurane anaesthetized horses [237] (23) Evaluation of the clinical efficacy of two partial intravenous anesthetic protocols, compared with isoflurane alone, to maintain general anesthesia in horses [238] (24) Continuous intravenous lidocaine infusion during isoflurane anaesthesia in horses undergoing surgical procedures [239] (25) A comparison of two morphine doses on the quality of recovery from general anaesthesia in horses [240] (26) Effects of a constant-rate infusion of dexmedetomidine on the minimal alveolar concentration of sevoflurane in ponies [241] (27) Comparison of the influence of two different constant-rate infusions (dexmedetomidine versus morphine) on anaesthetic requirements, cardiopulmonary function and recovery quality in isoflurane anaesthetized horses [242] (28) Cardiopulmonary effects of an infusion of remifentanil or morphine in horses anesthetized with isoflurane and dexmedetomidine [243] (29) Effects of a continuous rate infusion of butorphanol in isoflurane-anesthetized horses on cardiorespiratory parameters, recovery quality, gastrointestinal motility and serum cortisol concentrations [244] (30) Minimum end-tidal sevoflurane concentration necessary to prevent movement during a constant rate infusion of morphine, or morphine plus dexmedetomidine in ponies [245] (31) Effects of a constant rate infusion of medetomidine-propofol on isoflurane minimum alveolar concentrations in horses [246] (32) Cardiopulmonary effects and recovery quality of remifentanil–isoflurane anesthesia in horses [247] (33) Clinical comparison of two regimens of lidocaine infusion in horses undergoing laparotomy for colic [248] (34) Effects of medetomidine constant rate infusion on sevoflurane requirement, cardiopulmonary function, and recovery quality in Thoroughbred racehorses undergoing arthroscopic surgery [249] (35) The cardiovascular status of isoflurane-anaesthetized horses with and without dexmedetomidine constant rate infusion evaluated at equivalent depths of anaesthesia [250] (36) Cardiopulmonary effects and anaesthesia recovery quality in horses anaesthetized with isoflurane and low-dose S-ketamine or medetomidine infusions [251] (37) Comparison of the effects of an intravenous lidocaine infusion combined with 1% isoflurane versus 2% isoflurane alone on selected cardiovascular variables and recovery characteristics during equine general anaesthesia [252] (38) Clinical usefulness of intravenous constant rate infusion of fentanyl and medetomidine under sevoflurane anesthesia in Thoroughbred racehorses undergoing internal fixation surgery [253] (39) Effect of detomidine or romifidine constant rate infusion on plasma lactate concentration and inhalant requirements during isoflurane anaesthesia in horses [254] (40) Clinical comparison of dexmedetomidine and medetomidine for isoflurane balanced anaesthesia in horses [255] (41) Clinical evaluation of constant rate infusion of alfaxalone-medetomidine combined with sevoflurane anesthesia in Thoroughbred racehorses undergoing arthroscopic surgery [256] (42) Clinical effects of constant rate infusions of medetomidine-propofol combined with sevoflurane anesthesia in Thoroughbred racehorses undergoing arthroscopic surgery [257] (43) Plasma concentrations at two dexmedetomidine constant rate infusions in isoflurane anaesthetized horses: A clinical study [258]
2	(44) Hemodynamic function during neurectomy in halothane-anesthetized horses with or without constant dose detomidine infusion [259] (45) Combination of continuous intravenous infusion using a mixture of guaifenesin-ketamine-medetomidine and sevoflurane anesthesia in horses [260] (46) Evaluation of a mixture of thiopental-guafinesine-medetomidine and sevoflurane anesthesia in horses [261] (47) Influence of ketamine or xylazine supplementation on isoflurane anaesthetized horses--a controlled clinical trial [262] (48) Comparative study on sevoflurane anesthesia alone and combined with partial intravenous anesthesia using dexmedetomidine in healthy horses [263]
3	(49) Problems associated with perioperative morphine in horses: A retrospective case analysis [264] (50) Recovery quality after romifidine versus detomidine infusion during isoflurane anesthesia in horses [265]
4	(51) Minimal alveolar concentration of desflurane in combination with an infusion of medetomidine for the anaesthesia of ponies [266] (52) Influence of gastrointestinal tract disease on pharmacokinetics of lidocaine after intravenous infusion in anesthetized horses [267] (53) Clinical assessment of anesthesia with isoflurane and medetomidine in 300 equidae [268] (54) Partial intravenous anaesthesia in 5 horses using ketamine, lidocaine, medetomidine and halothane [269]
5	(55) Prolonged anesthesia using sevoflurane, remifentanil and dexmedetomidine in a horse [270] (56) Anesthetic management with sevoflurane combined with alfaxalone-medetomidine constant rate infusion in a Thoroughbred racehorse undergoing a long-time orthopedic surgery [271]
Total PIVA publications = 56	

LoE: level of evidence.

**Table 7 animals-11-01777-t007:** Total of publications giving information about effects on the recovery phase of: (7a) “*loco-regional*”, (7b) “*neuromuscular blocking agents (NMBAs)*” and (7c) “*other drugs*” given during the maintenance of general anaesthesia.

LoE and Topic	References
(7a) Loco-regional	
1	(1) The effect of epidural xylazine on halothane minimum alveolar concentration in ponies [272] (2) Epidural morphine and detomidine decreases postoperative hindlimb lameness in horses after bilateral stifle arthroscopy [273] (3) Castration of horses under total intravenous anaesthesia: Analgesic effects of lidocaine [274] (4) Local mepivacaine before castration of horses under medetomidine isoflurane balanced anaesthesia is effective to reduce perioperative nociception and cytokine release [275] (5) Intratesticular mepivacaine versus lidocaine in anaesthetised horses undergoing Henderson castration [276] (6) The effect of intra-articular mepivacaine administration prior to carpal arthroscopy on anesthesia management and recovery characteristics in horses [277]
2	/
3	(7) Epidural administration of opioid analgesics improves quality of recovery in horses anaesthetised for treatment of hindlimb synovial sepsis [278] (8) Clinical assessment of an ipsilateral cervical spinal nerve block for prosthetic laryngoplasty in anesthetized horses [279]
4	(9) The outcome of epidural anaesthesia in horses with perineal and tail melanomas: Complications associated with ataxia and the risks of rope recovery [280]
5	/
Total loco-regional publications = 9	
(7b) NMBAs	
1	(1) Influence of atracurium on cardiovascular parameters in horses undergoing vitrectomy during general anaesthesia, and on recovery duration and quality [281] (2) Effects of acetylcholinesterase inhibition on quality of recovery from isoflurane-induced anesthesia in horses [282]
2	/
3	/
4	(3) Neuromuscular and cardiovascular effects of atracurium in ponies anesthetized with halothane [283] (4) Neuromuscular and cardiovascular effects of atracurium administered to healthy horses anesthetized with halothane [284] (5) Clinical use of the neuromuscular blocking agents atracurium and pancuronium for equine anesthesia [285] (6) Effects of atracurium administered by continuous intravenous infusion in halothane-anesthetized horses [286] (7) Interaction of gentamycin and atracurium in anaesthetised horses [287] (8) Use of sevoflurane for anesthetic management of horses during thoracotomy [288]
5	(9) Prolonged neuromuscular blockade in a horse following concomitant use of vecuronium and atracurium [289]
?	(10) Biochemical effects of succinylcholine chloride in mechanically ventilated horses anesthetized with halothane in oxygen [290]
Total NMBAs publications = 10	
(7c) Other drugs	
1	(1) Effects of intravenous administration of dimethyl sulfoxide on cardiopulmonary and clinicopathologic variables in awake or halothane-anesthetized horses [291] (2) Effects of glycopyrrolate on cardiorespiratory function in horses anesthetized with halothane and xylazine [292] (3) Effects of a muscarinic type-2 antagonist on cardiorespiratory function and intestinal transit in horses anesthetized with halothane and xylazine [293] (4) Effect of dantrolene premedication on various cardiovascular and biochemical variables and the recovery in healthy isoflurane-anesthetized horses [294]
2	(5) Effects of dopamine, dobutamine, dopexamine, phenylephrine, and saline solution on intramuscular blood flow and other cardiopulmonary variables in halothane-anesthetized ponies [295]
3	(6) Doxapram infusion during halothane anaesthesia in ponies [296] (7) Influence of hypertonic saline solution 7.2% on different hematological parameters in awake and anaesthetized ponies [297]
4	(8) Temporal effects of an infusion of dopexamine hydrochloride in horses anesthetized with halothane [298]
5	/
Total other drugs publications = 8	

?: publications in which the study was not able to be classified according to LoE based on the available information.

**Table 8 animals-11-01777-t008:** Total of publications giving information about “*drugs before/during recovery*”.

LoE	References
1	(1) Antagonism of xylazine and ketamine anesthesia by 4-aminopyridine and yohimbine in geldings [299] (2) Effect of postoperative pethidine on the anaesthetic recovery period in the horse [300] (3) Recovery from sevoflurane anesthesia in horses: Comparison to isoflurane and effect of postmedication with xylazine [301] (4) Recovery phase of horses after general anesthesia with inhalants with and without postanesthetic sedation with xylazine (Rompun^®^) [302] (5) Effects of alpha-2 adrenoceptor agonists during recovery from isoflurane anaesthesia in horses [303] (6) Standing behavior in horses after inhalation anesthesia with isoflurane (Isoflo) and postanesthetic sedation with romifidine (Sedivet) or xylazine (Rompun) [304] (7) A comparison of equine recovery characteristics after isoflurane or isoflurane followed by a xylazine-ketamine infusion [305] (8) Effect of administration of propofol and xylazine hydrochloride on recovery of horses after four hours of anesthesia with desflurane [306] (9) Comparison of the effects of xylazine and romifidine administered perioperatively on the recovery of anesthetized horses [307] (10) Evaluation of infusions of xylazine with ketamine or propofol to modulate recovery following sevoflurane anesthesia in horses [308] (11) Effects of postanesthetic sedation with romifidine or xylazine on quality of recovery from isoflurane anesthesia in horses [309] (12) Assessment of unassisted recovery from repeated general isoflurane anesthesia in horses following post-anesthetic administration of xylazine or acepromazine or a combination of xylazine and ketamine [310] (13) Effect of postoperative xylazine administration on cardiopulmonary function and recovery quality after isoflurane anesthesia in horses [311] (14) Recovery from desflurane anesthesia in horses with and without post-anesthetic xylazine [312] (15) Comparison between the effects of postanesthetic xylazine and dexmedetomidine on characteristics of recovery from sevoflurane anesthesia in horses [313] (16) Pharmacokinetics and clinical effects of xylazine and dexmedetomidine in horses recovering from isoflurane anesthesia [314] (17) Recovery quality following a single post-anaesthetic dose of dexmedetomidine or romifidine in sevoflurane anaesthetised horses [315] (18) The effects of flumazenil on ventilatory and recovery characteristics in horses following midazolam-ketamine induction and isoflurane anaesthesia [316]
2	/
3	/
4	/
5	/
Total drugs before/during recovery publications = 18	

LoE: level of evidence.

**Table 9 animals-11-01777-t009:** Total of publications giving information about “*recovery systems*”.

LoE	References
1	(1) Cardiopulmonary function following anesthesia in horses experiencing hydro pool recovery versus padded stall recovery [317] (2) Cardiopulmonary function in horses during anesthetic recovery in a hydropool [318] (3) Comparison of recoveries from anesthesia of horses placed on a rapidly inflating-deflating air pillow or the floor of a padded stall [319] (4) Arterial oxygen tension and pulmonary ventilation in horses placed in the Anderson Sling suspension system after a period of lateral recumbency and anaesthetised with constant rate infusions of romifidine and ketamine [320] (5) Comparison between head-tail-rope assisted and unassisted recoveries in healthy horses undergoing general anesthesia for elective surgeries [321]
2	/
3	(6) Comparison of non-assisted versus head and tail rope-assisted recovery after emergency abdominal surgery in horses [322] (7) Effect of head and tail rope-assisted recovery of horses after elective and emergency surgery under general anaesthesia [323]
4	(8) Use of a pool-raft system for recovery of horses from general anesthesia: 393 horses (1984–2000) [324] (9) Use of a hydro-pool system to recover horses after general anesthesia: 60 cases [325] (10) Investigation into the assisted standing up procedure in horses during recovery phase after inhalation anaesthesia [326] (11) Use of the Anderson Sling suspension system for recovery of horses from general anesthesia [327] (12) Tilt table recovery of horses after orthopedic surgery: Fifty-four cases (1994–2005) [328] (13) Evaluation of a new full-body animal rescue and transportation sling in horses: 181 horses (1998–2006) [329] (14) Use of propofol-xylazine and the Anderson Sling Suspension System for recovery of horses from desflurane anesthesia [330] (15) Anaesthetic management for hydropool recovery in 50 horses [331] (16) A retrospective report (2003–2013) of the complications associated with the use of a one-man (head and tail) rope recovery system in horses following general anaesthesia [332]
5	(17) Internal fixation of a fractured axis in an adult horse [333] (18) Reduction and external coaptation as successful treatment for tarsocrural joint luxation in an Arabian mare [334]
Total recovery systems publications = 18	

LoE: level of evidence.

**Table 10 animals-11-01777-t010:** Total of publications giving information about “*respiratory system in recovery*”.

LoE	References
1	(1) Intranasal phenylephrine reduces post anesthetic upper airway obstruction in horses [335] (2) An evaluation of apnea or spontaneous ventilation in early recovery following mechanical ventilation in the anesthetized horse [336] (3) Cardiopulmonary effects associated with head-down position in halothane-anesthetized ponies with or without capnoperitoneum [337] (4) High inspired oxygen concentrations increase intrapulmonary shunt in anaesthetized horses [338] (5) Intermittent positive pressure ventilation with constant positive end-expiratory pressure and alveolar recruitment manoeuvre during inhalation anaesthesia in horses undergoing surgery for colic, and its influence on the early recovery period [339] (6) Influence on horse’s pulmonary function using a modified “Open-Lung-Concept”-Ventilation with different oxygen-concentration during general anaesthesia [340] (7) Effects of hypercapnic hyperpnea on recovery from isoflurane or sevoflurane anesthesia in horses [341] (8) Oxygenation and plasma endothelin-1 concentrations in healthy horses recovering from isoflurane anaesthesia administered with or without pulse-delivered inhaled nitric oxide [342] (9) Effect of pressure support ventilation during weaning on ventilation and oxygenation indices in healthy horses recovering from general anesthesia [343] (10) Comparison of cardiorespiratory variables in dorsally recumbent horses anesthetized with guaifenesin-ketamine-xylazine spontaneously breathing 50% or maximal oxygen concentrations [344] (11) Impact of low inspired oxygen fraction on oxygenation in clinical horses under general anesthesia [345] (12) The effect of two different intra-operative end-tidal carbon dioxide tensions on apnoeic duration in the recovery period in horses [346] (13) Controlled mechanical ventilation with constant positive end-expiratory pressure and alveolar recruitment manoeuvres during anaesthesia in laterally or dorsally recumbent horses [347] (14) Effect of reducing inspired oxygen concentration on oxygenation parameters during general anaesthesia in horses in lateral or dorsal recumbency [348] (15) Effects of 12 and 17 cm H_2_O positive end-expiratory pressure applied after alveolar recruitment maneuver on pulmonary gas exchange and compliance in isoflurane-anesthetized horses [349]
2	(16) Arterial blood gas tensions in the horse during recovery from anesthesia [350] (17) Alleviation of postanesthetic hypoxemia in the horse [351] (18) Restoration of arterial oxygen tension in horses recovering from general anaesthesia [352]
3	/
4	(19) Evaluation of a modification of the Hudson demand valve in ventilated and spontaneously breathing horses [353] (20) Effects of sedation, anesthesia, and endotracheal intubation on respiratory mechanics in adult horses [354] (21) Use of nasotracheal intubation during general anesthesia in two ponies with tracheal collapse [355]
5	/
Total respiratory system in recovery publications = 21	

LoE: level of evidence.

**Table 11 animals-11-01777-t011:** Total of publications giving information about the effects of “*other factors*” on the recovery phase.

Other Factors	LoE	References
Orthopaedic	1	(1) An in vitro biomechanical investigation of the mechanical properties of dynamic compression plated osteotomized adult equine tibiae [356] (2) Evaluation of a technique for collection of cancellous bone graft from the proximal humerus in horses [357]
	2	/
	3	/
	4	(3) Medial condylar fractures of the third metatarsal bone in horses [358] (4) Use of tension band wires in horses with fractures of the ulna: 22 cases (1980–1992) [359] (5) Application of a hook plate for management of equine ulnar fractures [360] (6) Fractures of the palmar aspect of the carpal bones in horses: 10 cases (1984–2000) [361] (7) A lateral approach for screw repair in lag fashion of spiral third metacarpal and metatarsal medial condylar fractures in horses [362] (8) Arthroscopic removal of discrete palmar carpal osteochondral fragments in horses: 25 cases (1999–2013) [363] (9) Surgical repair of propagating condylar fractures of the third metacarpal/metatarsal bones with cortical screws placed in lag fashion in 26 racehorses (2007–2015) [364]
	5	(10) A musculoskeletal model of the equine forelimb for determining surface stresses and strains in the humerus--part I. Mathematical modeling [365] (11) Successful reduction and internal fixation of an open tibial fracture in an adult Icelandic horse [366] (12) Internal fixation of a complete ventral luxation of the dens axis in an American quarter horse yearling [367]
Abdominal	1	(13) Incisional complications following exploratory celiotomy: Does an abdominal bandage reduce the risk? [368] (14) Case series evaluating the use of absorbable staples compared with metallic staples in equine ventral midline incisions [369]
	2	(15) Evaluation of iodophor skin preparation techniques and factors influencing drainage from ventral midline incisions in horses [370] (16) Pulmonary gas exchange and plasma lactate in horses with gastrointestinal disease undergoing emergency exploratory laparotomy: A comparison with an elective surgery horse population [371]
	3	(17) Comparison of laparoscopic versus conventional open cryptorchidectomies on intraoperative and postoperative complications and duration of surgery, anesthesia, and hospital stay in horses [372]
	4	(18) Investigation of perioperative and anesthetic variables affecting short-term survival of horses with small intestinal strangulating lesions [373] (19) Retrospective identification of bacterial isolates from emergency laparotomy surgical site infections in horses [374]
	5	/
Ocular surgery	1	/
	2	/
	3	(20) Complications associated with anaesthesia for ocular surgery: A retrospective study 1989–1996 [375] (21) Evaluation of risk factors, including fluconazole administration, for prolonged anesthetic recovery times in horses undergoing general anesthesia for ocular surgery: 81 cases (2006–2013) [376] (22) Factors associated with postoperative complications in healthy horses after general anesthesia for ophthalmic versus non-ophthalmic procedures: 556 cases (2012–2014) [377]
	4	(23) Transpalpebral enucleation using a chain écraseur [378]
	5	(24) Anesthesia case of the month. Evaluation and treatment of suspected squamous cell carcinoma of the third eyelid [379]
Others		
Airway	3	(25) Laryngoplasty with ventriculectomy or ventriculocordectomy in 104 draft horses (1992–2000) [380]
Body temperature	1	(26) Temporal changes in core body temperature in anesthetized adult horses [381]
	2	(27) Comparison of peripheral and core temperatures in anesthetized horses [382]
Repeated GA	2	(28) The effects of multiple anaesthetic episodes on equine recovery quality [383]
Environmental light	1	(29) Recovery of horses from general anesthesia in a darkened or illuminated recovery stall [384]
Cardiac	3	(30) Management and complications of anesthesia for transvenous electrical cardioversion of atrial fibrillation in horses: 62 cases (2002–2006) [385]
	4	(31) Hemodynamics before and after conversion of atrial-fibrillation to normal sinus rhythm in horses [386] (32) Electrocardiographic indicators of excitability in horses for predicting recovery quality after general anaesthesia [387]
Metabolic	1	(33) Effects of sevoflurane dose and mode of ventilation on cardiopulmonary function and blood biochemical variables in horses [388]
	2	(34) Influence of azaperone/metomidate in anesthesia on blood biochemistry in horse [389] (35) Ultrasonography of the equine triceps muscle before and after general anaesthesia and in post anaesthetic myopathy [390] (36) Metabolism during anaesthesia and recovery in colic and healthy horses: A microdialysis study [391]
	4	(37) Metabolic changes associated with anaesthesia and cherry poisoning in a pony [392] (38) The relationship of muscle perfusion and metabolism with cardiovascular variables before and after detomidine injection during propofol-ketamine anaesthesia in horses [393]
Miscellaneous	1	(39) A prospective clinical trial comparing metrizamide and iohexol for equine myelography [394]
	2	(40) Corneal abrasion and microbial contamination in horses following general anaesthesia for non-ocular surgery [395]
	4	(41) Electroacupuncture treatment for isoflurane induced hypotension in horses [396]
Total other factors publications = 41		

LoE: level of evidence. GA: general anaesthesia.

**Table 12 animals-11-01777-t012:** Total of: (12a) “*case series*” and (12b) “*case reports*” giving information about the recovery period, mainly complications that may happen during this phase.

**(12a) Case series, LoE = 4**	**References**
Aspiration pneumonitis	(1) Aspiration pneumonitis (Mendelson’s syndrome) as perianaesthetic complication occurring in two horses: A case report [397]
Behaviour during recovery from GA	(2) The behaviour of horses recovering from anaesthesia [398]
Guttural pouch mycosis	(3) Treatment of guttural pouch mycosis [399] (4) Potential for iatrogenic coil embolization of the caudal cerebellar artery during treatment of internal carotid artery bifurcation in two horses with guttural pouch mycosis [400]
Post-anaesthetic myopathy/neuropathy	(5) Post-anaesthetic forelimb lameness in horses [401] (6) Postanesthetic hind limb adductor myopathy in five horses [402] (7) Femoral nerve paralysis after general anaesthesia [403] (8) The incidence of post-anaesthetic myopathy with the use of a static air mattress [404] (9) Transient pelvic limb neuropathy following proximal metatarsal and tarsal magnetic resonance imaging in seven horses [405]
Pulmonary oedema	(10) Post-anesthetic pulmonary edema in two horses [406]
Repetitive injectable anaesthesia	(11) Repetitive injectable anesthesia in a 27-year-old horse [407]
Sweating	(12) Unexpected responses following intravenous pethidine injection in two horses [408]
Total case series = 12	
**(12b) Case reports, LoE = 5**	**References**
Assisted recovery with an Animal Rescue and Transport Sling (ARTS)	(1) Successful treatment of a coxofemoral luxation in a Shetland pony by closed reduction and prolonged immobilization using a full-body Animal Rescue Sling [409]
Bladder rupture	(2) Ruptured urinary bladder in a horse [410]
Cardiac arrest in recovery	(3) Uterine prolapse in a mare leading to metritis, systemic inflammatory response syndrome, septic shock and death [411] (4) Successful cardiopulmonary resuscitation in a sevoflurane anaesthetized horse that suffered cardiac arrest at recovery [412]
Coxofemoral luxation	(5) Coxofemoral luxation in a horse during recovery from general anaesthesia [413]
Diaphragmatic hernia/rupture	(6) Fatal diaphragmatic rupture during recovery from general anaesthesia in a Standardbred horse [414] (7) Use of continuous positive airway pressure (CPAP) in a horse with diaphragmatic hernia [415] (8) Perianesthetic development of diaphragmatic hernia in a horse with equine pituitary pars intermedia dysfunction (PPID) [416]
Epistaxis/Guttural pouch mycosis	(9) Unusual internal carotid artery branching that prevented arterial occlusion with a balloon-tipped catheter in a horse [417]
Facial paralysis	(10) Multimodal therapy including electroacupuncture for the treatment of facial nerve paralysis in a horse [418]
Incisional dehiscence	(11) Anesthetic management of an incisional dehiscence in recovery following exploratory laparotomy in a horse [419]
Laryngeal oedema	(12) Complications associated with the use of the cuffless endotracheal tube in the horse [420]
Laryngeal paralysis/dysfunction	(13) Temporary bilateral laryngeal paralysis in a horse associated with general anaesthesia and post anaesthetic myositis [421] (14) Acute life-threatening laryngeal dysfunction in a draft horse recovering from general anesthesia: A case report. [422]
Nasotracheal	(15) Broken nasotracheal tube aspiration in a horse during anaesthetic recovery [423]
Fractures	(16) Comparison of therapeutic techniques for the treatment of cheek teeth diseases in the horse: Extraction versus repulsion [424] (17) Limb fracture during recovery from general anaesthesia: An often tragic complication of equine anaesthesia [425] (18) Standing low-field magnetic resonance imaging of a comminuted central tarsal bone fracture in a horse [426]
Hyperthermia	(19*) Hyperthermia during isoflurane anaesthesia in a horse with suspected hyperkalemic periodic paralysis [427] (20*) Hyperthermia and delayed-onset myopathy after recovery from anesthesia in a horse [428]
Malignant hyperthermia	(21) Malignant hyperthermia in a halothane-anesthetized horse [429] (22*) Postanesthetic equine myopathy suggestive of malignant hyperthermia. A case report [430]
Post-anaesthetic myelomalacia	(23) Postanesthetic poliomyelomalacia in a horse [431] (24) Hemorrhagic myelomalacia following general anesthesia in a horse [432] (25) Hematomyelia in a colt: A post anesthesia surgery complication [433] (26) Post-anaesthetic myelomalacia in a horse [434]
Post-anaesthetic myelopathy	(27) Post-anaesthetic myelopathy in a 3-year-old Friesian gelding [435] (28) Post anaesthetic myelopathy in the horse: A case report [436]
Post-anaesthetic myopathy/neuropathy	(22*) Postanesthetic equine myopathy suggestive of malignant hyperthermia. A case report [430] (20*) Hyperthermia and delayed-onset myopathy after recovery from anesthesia in a horse [428] (29) Suspicion of postanesthetic femoral paralysis of the non-dependent limb in a horse [437]
Post-anaesthetic myositis	(30) Tracheal reconstruction by resection and end-to-end anastomosis in the horse [438]
Post-anaesthetic pleuropneumonia	(31) Pleuropneumonia as a sequela of myelography and general anaesthesia in a Thoroughbred colt [439]
Post-anaesthetic pulmonary haemorrhage	(32) Fatal post-anaesthetic pulmonary haemorrhage in a horse suffering from chronic-active exercise-induced pulmonary haemorrhage [440]
Prolonged recovery by HYPP	(33) Postanesthetic recumbency associated with hyperkalemic periodic paralysis in a quarter horse [441] (19*) Hyperthermia during isoflurane anesthesia in a horse with suspected hyperkalemic periodic paralysis [427]
Prolonged recovery by persistent hypoxaemia	(34) Prolonged recovery from general anesthesia possibly related to persistent hypoxemia in a draft horse [442]
Pulmonary oedema	(35) Negative pressure pulmonary edema as a post-anesthetic complication associated with upper airway obstruction in a horse [443] (36) Pulmonary oedema associated with anaesthesia for colic surgery in a horse [444] (37) Suspected air embolism associated with post-anesthetic pulmonary edema and neurologic sequelae in a horse [445] (38) Pulmonary edema at recovery after colic operation with in-situ nasogastric tube in a horse [446]
Tracheal rupture	(39) Tracheal rupture following general anaesthesia in a horse [447]
Total case reports = 39	

Abbreviations in alphabetical order: GA, general anaesthesia; HYPP, hyperkaelemic periodic paralysis. *: “case reports” present in two different categories within the same section.

**Table 13 animals-11-01777-t013:** Total of publications giving information about “*systems to score recoveries*”.

LoE	References
1	(1) A comparison of four systems for scoring recovery quality after general anaesthesia in horses [448] (2) Quantitative and qualitative comparison of three scoring systems for assessing recovery quality after general anaesthesia in horses [449] (3) Factors affecting the perception of recovery quality in horses after anaesthesia [450]
2	(4) A study of the correlation between objective and subjective indices of recovery quality after inhalation anaesthesia in equids [451] (5) Assessment of agreement among diplomates of the American College of Veterinary Anesthesia and Analgesia for scoring the recovery of horses from anesthesia by use of subjective grading scales and development of a system for evaluation of the recovery of horses from anesthesia by use of accelerometry [452]
3	(6) Retrospective evaluation of correlation and agreement between two recovery scoring systems in horses [453]
4	/
5	/
Total systems to score recoveries publications = 6	

LoE: level of evidence.

## Data Availability

Not applicable.

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
