# Peer review of "Recovery after General Anaesthesia in Adult Horses: A Structured Summary of the Literature"

_animals, 2021, doi:10.3390/ani11061777_

Round 1

Reviewer 1 Report

The manuscript provides a comprehensive review of the literature related to anaesthetic recovery, a highly relevant topic in equine anaesthesia. Since many factors are involved, and contradictory data have been observed in some cases, such an integrated review is of high interest for the reader, either specialist or clinical practitioner performing equine anaesthesia.

As stated by the authors, a scoping review has been followed with a comprehensive review of the literature using two known databases (Pubmed and WOS) in a structured way, and assigning the level of evidence of the found references. A systematic review would have not been feasible considering such a broad question “Recovery from anaesthesia in horses”. However, the relatively simple, and broad, search strategy failed to identify some references and an alternative search should have been performed in order to identify them or, at least, to ensure relevant references were not missed. Actually, the authors have considered the advantages of Mesh terms for Pubmed searches, but not in the current manuscript search. Since the first author’s aim was to determine “How many studies have been published on this topic until today?“ It would be helpful a brief discussion on the potential, or not, relevance of those references not found and indicate how the authors identified additional references, already included in the manuscript, but not initially found.

The review is overall well written and clear. References have been structured in meaningful areas related to recovery.

Title: Would the year be enough for the reader to understand when the review was performed with updated references? I do not think the title should consider the day and month: this is already included in the M&M section. Terminology is still confusing for the reviewer: although a structured review is claimed, the PRISMA-ScR guidelines for scoping reviews were followed. Were all guidelines items followed, when applicable, to consider this manuscript a Scoping Review?

M&M

The scientific evidence of each study has been clearly defined. There were discrepancies between authors when assigning LoE or this process was straightforward?

Results

Table 3 is too long and although useful to identify selected references with titles and the associated objective of meaningful results, it may be considered as supplemental material to be downloaded if required.

Discussion

The grouping approach, although somewhat heterogeneous, allows a consistent inclusion of references and data provided. This part is useful for the reader interested in the recovery phase after general anaesthesia in adult horses.

Line 795: The reviewer acknowledges this is not a systematic review where a more complex search strategy would have been considered but to help determine whether a systematic review of the literature is warranted. As already indicated above, some relevant papers were not found with the search strategy employed (references 1,464,466,468,479-481 but also 471-473, 476). It is unclear from the reviewer how these references were identified and whether other relevant papers might be missing. Although it is highly likely the authors have not missed relevant references, they should ensure actually no relevant references are missed. Please explain briefly.

Conclusions: Three aims were considered. However, the conclusions are mostly related to the third aim but not the amount of data available (first) or the evidence (LoE) these data/references provide (second). Considering all aspects and factors related to recovery, it becomes difficult to provide a general approach. However, are the conclusions provided actually based on a high LoE? For example, is the current tendency towards sedation as an alternative to anaesthesia well supported by LoE. What recommendations can be followed based on high LoE?

Minor comments:

Line 611: 15 “L/min”….?

Line 645: “fractures died” or actually “horses with fractures died”?

Author Response

Dear reviewer

Thank you very much for your positive feed-back and for helping us improving our manuscript. All your suggestions have been addressed. Please find our comments included in the text below.

Sincerely

Answers to comments:

As stated by the authors, a scoping review has been followed with a comprehensive review of the literature using two known databases (Pubmed and WOS) in a structured way, and assigning the level of evidence of the found references. A systematic review would have not been feasible considering such a broad question “Recovery from anaesthesia in horses”. However, the relatively simple, and broad, search strategy failed to identify some references and an alternative search should have been performed in order to identify them or, at least, to ensure relevant references were not missed. Actually, the authors have considered the advantages of Mesh terms for Pubmed searches, but not in the current manuscript search. Since the first author’s aim was to determine “How many studies have been published on this topic until today?“ It would be helpful a brief discussion on the potential, or not, relevance of those references not found and indicate how the authors identified additional references, already included in the manuscript, but not initially found.

Thank you very much for pointing this out; the topic is addressed further on in the discussion section.

The review is overall well written and clear. References have been structured in meaningful areas related to recovery.

Title: Would the year be enough for the reader to understand when the review was performed with updated references? I do not think the title should consider the day and month: this is already included in the M&M section. Terminology is still confusing for the reviewer: although a structured review is claimed, the PRISMA-ScR guidelines for scoping reviews were followed. Were all guidelines items followed, when applicable, to consider this manuscript a Scoping Review?

The title has been shortened as suggested by the reviewer.

Regarding terminology/PRISMA guidelines: for this structured review we could not follow the classic guideline/checklist for typical systematic reviews (see attachment PRISMA 2009 Systematic & Metanalyses checklist) which analyze the data of the studies included same as in a meta-analysis. Therefore, and after consulting an expert in reporting guidelines (Dr. Jennifer de Beyer, University of Oxford, at the AVA online Scientific Writing workshop, 10 – 11 April 2021), the immediate suggestion was to use the PRISMA-ScR guidelines (see Figure S1 of the manuscript), as we are not analyzing results of the studies included, we rather only refer to them and classify them. This ScR version of PRISMA guidelines does not include sections such as: “Present results of each meta-analysis done, including confidence intervals and measures of consistency”, for instance. All items of the PRISMA-ScR guidelines were followed.

M&M

The scientific evidence of each study has been clearly defined. There were discrepancies between authors when assigning LoE or this process was straightforward?

The process was straightforward as LoE were clearly defined by table 1 in advance (as described in Table 1).

Results

Table 3 is too long and although useful to identify selected references with titles and the associated objective of meaningful results, it may be considered as supplemental material to be downloaded if required.

All the tables summarizing the included articles (tables 3 – 13) are equally important. Therefore, we would prefer to leave all the tables in the main manuscript.

Discussion

The grouping approach, although somewhat heterogeneous, allows a consistent inclusion of references and data provided. This part is useful for the reader interested in the recovery phase after general anaesthesia in adult horses.

Line 795: The reviewer acknowledges this is not a systematic review where a more complex search strategy would have been considered but to help determine whether a systematic review of the literature is warranted. As already indicated above, some relevant papers were not found with the search strategy employed (references 1,464,466,468,479-481 but also 471-473, 476). It is unclear from the reviewer how these references were identified and whether other relevant papers might be missing. Although it is highly likely the authors have not missed relevant references, they should ensure actually no relevant references are missed. Please explain briefly.

Additional information regarding “missing” papers has been added to the discussion section, see Lines 860-864: No systematic approach was followed in order to find articles not detected by the structured reviewing process. However, the present discussion was written by two experts in the field and by doing a literature search as usual to write a scientific manuscript. Therefore, it is unlikely that important articles were overlooked when discussing the results.

Conclusions: Three aims were considered. However, the conclusions are mostly related to the third aim but not the amount of data available (first) or the evidence (LoE) these data/references provide (second). Considering all aspects and factors related to recovery, it becomes difficult to provide a general approach. However, are the conclusions provided actually based on a high LoE? For example, is the current tendency towards sedation as an alternative to anaesthesia well supported by LoE. What recommendations can be followed based on high LoE?

The goal of a structured review is to minimize subjectivity, therefore data is presented as objectively as possible. However, Scientific evidence was taken into account when summarizing the results. This information has been added to the manuscript L224-225.

Minor comments:

Line 611: 15 “L/min”….?

Information added, thank you.(L: 658)

Line 645: “fractures died” or actually “horses with fractures died”?

Changed, thank you very much. (L: 692)

Reviewer 2 Report

Dear Authors,

Thank you for a very well written structured review of this important aspect of equine anaesthesia. I am familiar with this manuscript and my previous concerns have already been addressed. This version is really good and complete. This information will help guide future studies on where we should investigate next.

Well done, I don't have any minor or major edits on this version. I was wondering if this journal will accept all of the tables in the results, or if text results should be summarised and the tables should be referred to! This will be up to the editorial team I would guess. 

Well done,

Reviewer

Author Response

Dear reviewer

Thank you very much for you positive feed-back and for all the work you did put into our manuscript.

Sincerely